# OSWorld-MCP: Benchmarking MCP Tool Invocation In Computer-Use Agents

**Hongrui Jia**[1,2,†], **Jitong Liao**[2,†], **Xi Zhang**[2,†], **Haiyang Xu**[2,*], **Tianbao Xie**[2], **Chaoya Jiang**[1], **Ming Yan**[2,*], **Si Liu**[3], **Wei Ye**[1,*], **Fei Huang**[2]
[1] Peking University    [2] Tongyi Lab, Alibaba Group    [3] Beijing Zhongguancun Academy
{jiahongrui, wye}@pku.edu.cn, shuofeng.xhy@alibaba-inc.com

## ABSTRACT

With advances in decision-making and reasoning capabilities, multimodal agents show strong potential in computer application scenarios. Past evaluations have mainly assessed GUI interaction skills, while tool invocation abilities, such as those enabled by the Model Context Protocol (MCP), have been largely overlooked. Comparing agents with integrated tool invocation to those evaluated only on GUI interaction is inherently unfair. We present OSWorld-MCP, the first comprehensive and fair benchmark for assessing computer-use agents' tool invocation, GUI operation, and decision-making abilities in a real-world environment. We design a novel automated code-generation pipeline to create tools and combine them with a curated selection from existing tools. Rigorous manual validation yields 158 high-quality tools (covering 7 common applications), each verified for correct functionality, practical applicability, and versatility. Extensive evaluations of state-of-the-art multimodal agents on OSWorld-MCP show that MCP tools generally improve task success rates (e.g., from 8.3% to 17.6% for OpenAI o3 at 15 steps, from 38.9% to 45.0% for Claude 4 Sonnet at 50 steps), underscoring the importance of assessing tool invocation capabilities. However, even the strongest models have relatively low tool invocation rates, Only 33.3%, indicating room for improvement and highlighting the benchmark's challenge. By explicitly measuring MCP tool usage skills, OSWorld-MCP deepens understanding of multimodal agents and sets a new standard for evaluating performance in complex, tool-assisted environments. Our code, environment, and data are publicly available at https://osworld-mcp.github.io.

## 1 INTRODUCTION

Large Language Models (LLMs) such as GPT-5 (OpenAI, 2025a), DeepSeek-R1 (Guo et al., 2025a), and Qwen3 (Yang et al., 2025) have dramatically advanced reasoning and decision-making capabilities. Building on these advances, recent Large Multimodal Models (LMMs) are able to address complex computer-use tasks, which has stimulated considerable research interest (Qin et al., 2025; Lai et al., 2025; Song et al., 2025; Luo et al., 2025; Lu et al., 2025a;b; Ye et al., 2025; Wang et al., 2024b; 2025a; Zhu et al., 2025; Wang et al., 2024a). Consequently, how to reliably and robustly benchmark different LMMs in GUI-driven scenarios has become a key open question. Existing evaluation frameworks (Xie et al., 2024; Abhyankar et al., 2025; Xie et al., 2025; Bonatti et al., 2024; Kuntz et al., 2025; Rawles et al., 2024) primarily focus on assessing a model's ability to perform GUI-based operations, by predefining a set of user-interface actions (*e.g., click, type*, and *drag*) and allowing the model to autonomously decide how to complete the task.

Although many benchmarks have been proposed for evaluating GUI agents, most neglect a crucial capability: the ability to invoke external tools such as the Model Context Protocol (MCP) (Anthropic, 2024). MCP is an open standard that connects AI applications with external systems. By using MCP, computer-use agents can access diverse resources, including data sources such as local files and databases, and tools such as search engines and calculator, thereby enabling them to obtain critical information and complete tasks more effectively. In many cases, performing a task through MCP is more efficient than relying solely on GUI operations. For instance, as illustrated in Figure 1, when an agent is instructed to install the autoDocstring extension in VS Code, a GUI-based

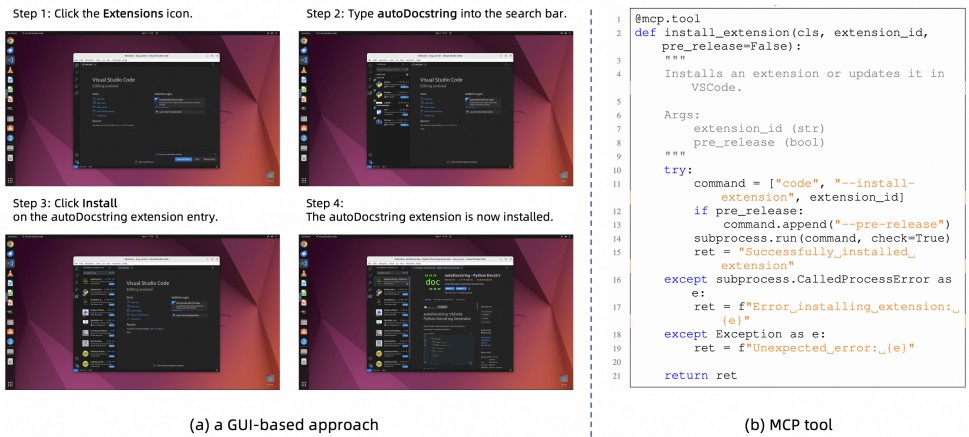

Figure 1: Comparison of completing the instruction *"Please help me install the autoDocstring extension in VS Code"* via GUI operations (a) and the MCP Tool (b).

approach may require at least four steps, whereas an MCP tool can achieve the same result in a single step, offering both greater efficiency and higher robustness. Several recent agents (Lai et al., 2025; Song et al., 2025) have already incorporated autonomous tool invocation and have achieved notable performance gains. However, it is inherently inequitable to compare such agents with others that assess only GUI interaction. At present, there remains a lack of comprehensive and equitable benchmarks that jointly evaluate GUI operation skills, tool invocation capabilities, and the decision-making competence of computer-use agents in an integrated framework.

To bridge this gap, we introduce OSWorld-MCP, the first comprehensive and fair benchmark designed to evaluate the tool invocation capabilities of computer-use agents.Our primary motivation is to establish a unified standard for fair comparison of tool utilization abilities across different models, addressing the current lack of consistency in tool sets and evaluation metrics. Built upon a widely used real-world computer-use environment OSWorld (Xie et al., 2024), OSWorld-MCP significantly extends its capabilities by incorporating a curated set of **158 high-quality MCP tools**. These tools cover **7** common applications such as LibreOffice Writer and VS Code, ensuring a diverse and realistic testing environment (Figure 4). A number of 25 non-target tools are included, serving as distractors in the OSWorld-MCP tasks. Besides, our designed MCP tools are applicable to 250 tasks, accounting for 69% of the entire benchmark, which underscores their broad applicability. Notably, 153 tasks, representing 42% of the benchmark, involve challenging multi-round tool invocations. Even the strongest model, Claude 4 Sonnet, achieves an accuracy of 0 when relying solely on GUI operations for tasks requiring four rounds of tool invocation, and only 19.5% accuracy when MCP tools are introduced. These results highlight the challenging of our benchmark and the broad utility of our tools in diverse scenarios.

Another distinguishing feature of OSWorld-MCP is its dynamic interaction between GUI operations and tool usage. Specifically, at every step of a task, the agent can autonomously choose between our MCP tools and direct GUI actions (*e.g., click* and *type*) to interact with the graphical interface. With this setting, OSWorld-MCP can provide a balanced and thorough assessment of LMM capabilities in hybrid decision-making skills. Here, the decision-making involves not only selecting the correct tools, but also choosing the most efficient execution path when both GUI operations and MCP tools are required to accomplish the task. Besides, to provide a more nuanced evaluation, we introduce two new metrics alongside task accuracy: *Tool Invocation Rate* (TIR) and *Average Completion Steps* (ACS). TIR measures the proportion of tasks successfully completed using MCP tools, offering insights into an agent's tool utilization propensity, while ACS quantifies task completion efficiency. In conclusion, compared with existing text-based tool-use benchmarks (Liu et al., 2025; Mo et al., 2025; Gao et al., 2025; Wang et al., 2025b) and the above GUI relevant benchmarks, OSWorld-MCP has significant advantages in evaluating real-world computer-use scenarios. It challenges agents to interpret visual GUI information, perform GUI operations, invoke appropriate tools, and effectively chain multiple tools. This combination of features makes OSWorld-MCP a more comprehensive and realistic benchmark for evaluating computer-use capabilities.

To develop the above-mentioned high-quality MCP tools, we design an automated code generation pipeline comprising three modules: the Code Generation Module, the Code Filter Module, and the Tool Wrap Module. By leveraging the advanced reasoning capabilities of OpenAI o3 (OpenAI, 2025b), this pipeline produces 72 functional tools. These tools are then combined with those curated from existing MCP servers (Lai et al., 2025), followed by a fine-grained manual verification procedure to remove functionally redundant items and highly task-specific ones that lack relevance to real-world applications. Through this process, we obtain a curated collection of 158 high-quality tools, each verified to be readily usable and aligned with the designed task difficulty. With this comprehensive tool set and our OSWorld-MCP, we conduct a detailed experimental study and analysis on a range of state-of-the-art LMMs and multimodal frameworks. Three key findings are identified during experiments: (1) **MCP tools enhance agent accuracy and efficiency compared to the GUI-only setting.** For example, the success rate of OpenAI o3 increases from 8.3% to 17.6% at 15 steps. (2) **The tool invocation rate positively correlates with performance.** We also observe that tool invocation rates for multimodal agents remain relatively low, indicating the significant potential in the tool utilization capabilities of current LMMs and multimodal frameworks. (3) **The composition of multiple tools remains a significant challenge.** Performance declines on complex task involving more tools. In the other hands, agent struggle on selecting the correct tool from a full list.

Our contributions are as follows:

- We introduce OSWorld-MCP, a comprehensive, fair, and novel benchmark for evaluating computer-use agents that integrates 158 high-quality MCP tools (covering 7 common apps) with GUI operations in real-world scenarios. It bridges the gap between pure-GUI and text-based tool-use evaluations, offering a holistic and realistic assessment of computer-use capabilities.
- We propose a novel pipeline combining automated code generation with rigorous manual curation to create MCP tools, enhancing our benchmark's evaluation depth and fairness. We also introduce new metrics (*i.e.*, TIR and ACS) for nuanced assessment of agents' tool utilization propensity.
- Our extensive experiments identify three key findings: (1) the effective use of MCP tools significantly improves the accuracy of LMMs and reduces the ACS; (2) the TIR is positively correlated with accuracy; (3) while MCP tools consistently enhance efficiency and accuracy as task difficulty increases, combining MCP tools remains more challenging than combining GUI operations.

## 2 RELATED WORK

### 2.1 BENCHMARKS FOR MULTIMODAL AGENTS

Current benchmarks (Deng et al., 2023; Lù et al., 2024; Kapoor et al., 2024; Zhou et al., 2023; Koh et al., 2024; Drouin et al., 2024; Tian et al., 2024; Bonatti et al., 2024; Xie et al., 2024) for multimodal agents primarily focus on evaluating their ability to complete tasks through GUI-based operations. Static benchmarks such as Mind2Web (Deng et al., 2023), WebLinx (Lù et al., 2024), and OmniAct (Kapoor et al., 2024) rely on manually collected static datasets to assess agent performance. These static benchmarks are built upon pre-defined trajectories of GUI actions, which make them incapable of evaluating alternative action paths that may arise when tool invocation is introduced. In contrast, dynamic interactive benchmarks operate in open-ended environments and provide reward signals upon task completion, enabling a more flexible and open-ended evaluation of agents. Notable examples of dynamic benchmarks for specific environments include WebArena (Zhou et al., 2023), VisualWebArena (Koh et al., 2024), WorkArena (Drouin et al., 2024), MMInA (Tian et al., 2024), WindowsAgentArena (Bonatti et al., 2024), and OSWorld (Xie et al., 2024). However, existing dynamic interactive benchmarks typically predefine only GUI actions for the agent to use, and therefore lack a comprehensive and fair evaluation framework that jointly measures multimodal agents' tool invocation, GUI interaction, and decision-making capabilities.

### 2.2 MODEL CONTEXT PROTOCOL

With the rapid development of multimodal agents, increasing attention has been paid to their tool invocation capabilities (Song et al., 2025; Lai et al., 2025). Introduced by Anthropic in November 2024, the Model Context Protocol (MCP) is a JSON-RPC–based client–server interface designed for secure context ingestion and structured tool invocation. MCP provides a standardized, model-agnostic interface that enables AI applications to connect to external systems such as tools, data

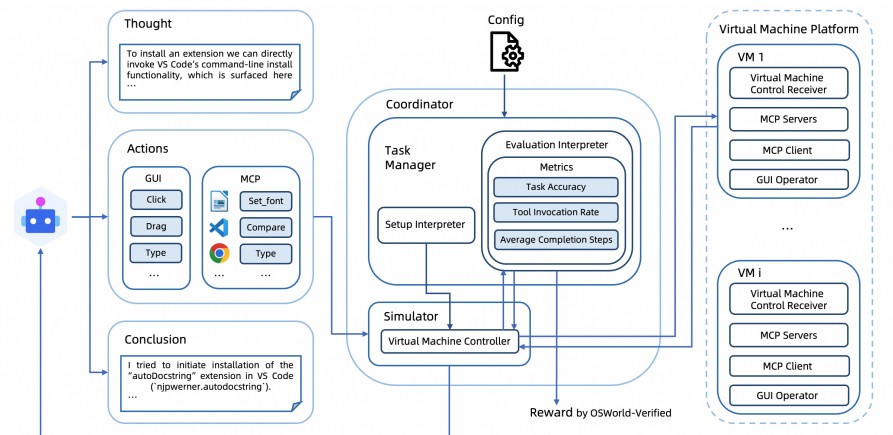

Figure 2: Overview of the OSWorld-MCP framework.

resources, and workflows, thereby facilitating the integration of large language models with external data sources and utilities (Anthropic, 2024). This standardization addresses challenges arising from fragmented and highly customized integrations. MCP supports flexible plug-and-play tooling, secure infrastructure integration, and cross-LLM vendor compatibility (Ehtesham et al., 2025). Several contemporary MCP-related benchmarks (Gao et al., 2025; Liu et al., 2025; Mo et al., 2025; Wang et al., 2025b). MCPEval (Liu et al., 2025) for LLM evaluation have recently emerged and MCP-Radar (Gao et al., 2025) cover only a limited set of MCP servers, typically no more than a few dozen tools, which restricts task diversity. LiveMCPBench (Mo et al., 2025) employs large language model–based evaluation, an approach that is not well suited to tasks requiring real-time knowledge. MCP-Bench (Wang et al., 2025b) defines its tasks based on available tools, introducing constraints that create a gap between benchmark tasks and truly open-ended real-world problems. Currently, there is no benchmark that comprehensively evaluates multimodal agents in terms of GUI interaction, tool invocation, and decision-making capabilities within an integrated and fair framework.

## 3  OSWORLD-MCP BENCH

### 3.1  OVERVIEW

We propose OSWorld-MCP, a comprehensive benchmark for evaluating computer-use agents. OSWorld-MCP is built upon the OSWorld benchmark, which is a widely used dynamic and interactive evaluation framework designed to assess the performance of multimodal agents in realistic computing environments, including Ubuntu, Windows, and macOS. OSWorld covers nine applications and consists of a total of 369 real-world tasks that involve interaction through both graphical user interfaces (GUI) and command-line interfaces (CLI). As illustrated in Figure 2, OSWorld-MCP enables effective assessment of multimodal agents in authentic scenarios, capturing their tool invocation capability, GUI operation skills, and decision-making competence. Decision-making assessment includes the ability to choose between GUI and MCP pathways, as well as the ability to select the most appropriate MCP tool for a given task. In the following section, we first introduce our automated pipeline for tool generation and the procedures for collecting and filtering tools used to construct OSWorld-MCP. We then present a detailed analysis of the MCP tools we produce, demonstrating the high quality and rational design of the OSWorld-MCP tool set. Finally, we describe the evaluation metrics defined for the OSWorld-MCP dataset and outline how these metrics enable comprehensive performance measurement of different models.

### 3.2  TOOLS GENERATION AND FILTER

Existing MCP servers generally offer tools that are relatively simple and often have overlapping functionalities. Consequently, there is a shortage of high-quality MCP tools that can be readily applied in practical scenarios. To address this limitation, we design an automated tool generation pipeline composed of three modules: the Code Generation Module, the Code Filter Module, and the

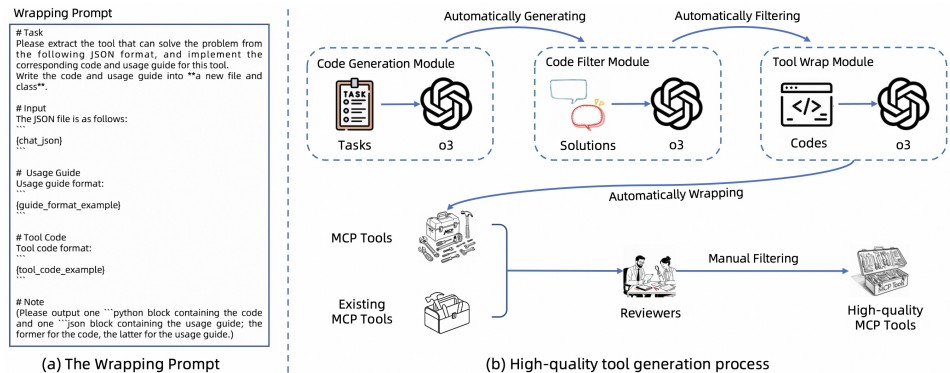

Figure 3: Illustration of our tool generation process. (a) Prompt for wrapping code into MCP tools with OpenAI o3. (b) Workflow for generating high-quality tools.

Tool Wrap Module. In the Code Generation Module, the user only needs to specify the target task, and the module automatically generates code to accomplish it. Specifically, we employ OpenAI o3 (OpenAI, 2025b) to produce code-based solutions for every task in OSWorld. Following the prompting strategy of CoAct (Song et al., 2025), we develop our own prompt and instruct OpenAI o3 to generate, whenever feasible, functional code solutions capable of completing the tasks. In the Code Filter Module, we use OpenAI o3 to summarize usable code obtained from multi-turn interactions. This summarized code is then applied to solve the corresponding tasks, and any code successfully completing the tasks is retained. Through this process, we obtain seventy-two verified solutions. In the Tool Wrap Module, we provide OpenAI o3 with a prompt that automates the packaging of these verified code solutions into 72 MCP tools, as illustrated in Figure 3.(a).

In addition, we carefully curate 192 tools from existing MCP servers. Since some generate or imported tools are tailored to solve a single specific task and thus unsuitable for real-world use, we conduct a manual inspection of all 264 collected tools to remove such task-specific items as well as functionality duplicates. Each tool is independently evaluated by at least two reviewers with extensive GUI agent development experience, and is retained only if both reviewers deemed it qualified. After this manual filtering process, we obtain 158 high-quality tools that are both applicable and valuable in real-world computing environments.

## 3.3 TOOLS ANALYSIS

We analyze the composition of the 158 high-quality tools designed for practical real-world use, with their distribution across application scenarios shown in Figure 4.(b). To further ensure that these tools have a tangible positive impact on task completion in realistic settings, we conduct an additional manual validation and find that 133 tools effectively contribute to improving task efficiency, while the remaining 25 tools originate from existing external MCP servers.

To verify that the effective tools can be actively utilized by models, we evaluate five state-of-the-art large multimodal models, including Qwen2.5-VL-72B-Instruct (Bai et al., 2025) and Claude 4 Sonnet (Anthropic, 2025), on OSWorld-MCP, allowing them at each step to autonomously choose between performing GUI operations and invoking any of the 158 high-quality tools. The results indicate that 131 tools are invoked at least once during evaluation. The remaining two tools, which are manually re-verified for usability, are hypothesized to be absent from model usage due to the complexity of the associated tasks, which likely discouraged models from attempting to invoke them. Tool invocation frequencies are presented in Figure 4.(a).

We also conduct a manual analysis of all 361 OSWorld-MCP tasks (8 Google Drive tasks excluded) to annotate the set of available tools for each task and to record the total number of invocations of these tools across evaluations. The distribution of total available tool invocations per task is shown in Figure 4.(c). An available tool is defined as one whose invocation can make task execution substantially more efficient. Based on this definition, OSWorld-MCP is classified into two subsets: **Tool-Beneficial** Tasks, which include tasks for which at least one available tool can improve effi-

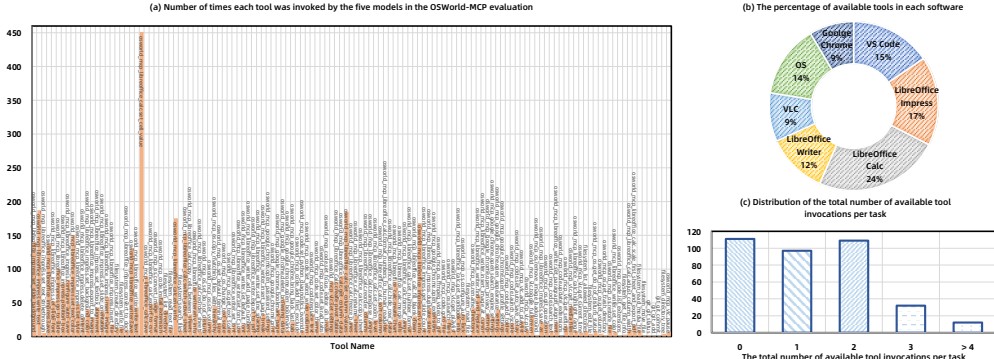

Figure 4: (a) illustrates the total number of times each tool was invoked by the five models in a single OSWorld-MCP evaluation. (b) depicts the distribution of our 158 high-quality tools across different usage scenarios for various OSWorld environments. Due to software version constraints within OSWorld, MCP servers were not developed for GIMP and Thunderbird. (c) presents the distribution of the total available tool invocations per task.

ciency (250 tasks), and **Non-Tool-Beneficial** Tasks, which include tasks for which no available tool improves efficiency (111 tasks).

These findings demonstrate that the tools in OSWorld-MCP are genuinely relevant to real-world needs and are not artificially tailored for specific benchmark tasks. Furthermore, many tasks can be completed more efficiently through multiple tool invocations, requiring models to select appropriate tools and to exhibit strong decision-making capabilities. This confirms the soundness of our tool design and underscores the challenging nature of the benchmark.

## 3.4 METRICS

Building on the evaluation of GUI operation capabilities, we introduce three metrics in OSWorld-MCP to assess a multimodal agent's tool invocation and decision-making abilities: Task Accuracy, Tool Invocation Rate, and Average Completion Steps.

**Task Accuracy.** Similar to OSWorld, we use accuracy as an overall performance indicator to measure how well a multimodal agent completes tasks. This metric jointly reflects the agent's decision-making ability, tool invocation skills, and GUI interaction performance.

**Tool Invocation Rate (TIR).** As described in Section 3.2, human reviewers classify each task into one of two categories: Tool-Beneficial Tasks or Non-Tool-Beneficial Tasks. Let $N_t$ be the total number of Tool-Beneficial Tasks, and $n_t$ the number of such tasks in which the agent invoked a tool and successfully completed the task during evaluation. Let $N_g$ be the total number of Non-Tool-Beneficial Tasks, and $n_g$ the number of such tasks in which the agent did not invoke a tool and successfully completed the task. We define TIR as:

$$\text{TIR} = (n_t + n_g)/(N_t + N_g) \tag{1}$$

When computing TIR within the Tool-Beneficial Task subset ($N_g = 0$), the formula reduces to TIR $= n_t/N_t$, which measures the proportion of Tool-Beneficial Tasks where the agent correctly invoked a tool and succeeded. When computing TIR within the Non-Tool-Beneficial Task subset ($N_t = 0$), the formula reduces to TIR $= n_g/N_g$, which measures the proportion of Non-Tool-Beneficial Tasks where the agent correctly avoided tool invocation and succeeded. TIR can effectively indicate the agent's ability to decide when a tool should or should not be invoked.

**Average Completion Steps (ACS).** This metric measures the average number of steps an agent takes to complete a task. For $N$ tasks, if the number of execution steps for task $i$ is $S_i$, the Average Completion Steps is computed as:

$$\text{ACS} = \sum_{i=1}^{N} S_i/N \tag{2}$$

ACS reflects decision-making efficiency: the more accurate the decisions, the higher the rate of correct tool usage, and the more frequently the agent selects efficient tools, the lower ACS will be.

# 4 EXPERIMENTS

## 4.1 SETUP

We evaluate a series of state-of-the-art Large Multimodal Models (LMMs), including Qwen2.5-VL-72B-Instruct (Bai et al., 2025), Qwen3-VL-Plus (QwenTeam, 2025), Seed1.5-VL (Guo et al., 2025b), Claude 4 Sonnet (Anthropic, 2025), OpenAI o3 (OpenAI, 2025b), and Gemini-2.5-Pro (Comanici et al., 2025), by running a computer-use agent on real-world tasks in OSWorld-MCP involving nine different software applications. To facilitate comparison of performance differences across models, we standardized our evaluation using the GUI-Owl agent configuration. This may lead to some performance fluctuations for certain models under the original OSWorld configuration. Specifically, at each step, the core LMM performs visual perception of the current interface, generates a corresponding thought, and proposes the next action along with a reasoning summary. This reasoning history informs subsequent planning during task execution. Besides, we also evaluate a multi-agent framework Agent-S2.5 (Simular Research, 2025), and adopt OpenAI o3 as the main generation model with UI-TARS-1.5-72B as the grounding model. For each task, the agent is restricted to a fixed maximum number of steps to either complete the task or determine that it is infeasible. The temperature parameter is set to 1.0.

In the original OSWorld configuration, LMMs may only use a predefined set of 11 basic GUI operations, including *key, type, mouse_move, click, drag, right_click, middle_click, double_click, scroll, wait and terminate*, to complete tasks. With the introduction of MCP tools in our OSWorld-MCP, the LMM can, autonomously decide whether to invoke any MCP Tool or perform a GUI operation at each action step. In details, we employ the 158 high-quality MCP tools curated in Section 3.2. Since providing all 158 tools simultaneously would result in excessively long input contexts, we apply Retrieval-Augmented Generation (RAG) to select only the tools relevant to the current application. The model then chooses from these filtered MCP tools or GUI operations.

## 4.2 MAIN RESULTS

As shown in Table 1, we evaluate six advanced end-to-end models and one agent-based frameworks under both the original OSWorld configuration and OSWorld-MCP with our curated high-quality tools, using maximum step limits of 15 and 50 respectively. Due to fluctuations in the experimental results, we conducted three runs for each model or framework under each configuration. The results reported in Tables 1 are the averages over these three runs.

Among the end-to-end models, Claude 4 Sonnet achieves the highest accuracy in OSWorld-MCP for both step limits, reaching 36.1 (15 steps) and 45.0 (50 steps) respectively among LMMs. Claude 4 Sonnet also records the highest tool invocation rate in OSWorld-MCP among these models. Qwen3-VL-PLUS achieve the lowest average completion steps (ACS) at 15 and 50 steps respectively, with values of 10.0 and 18.6. Our results also reveal that current large multimodal models generally have low tool invocation rates in OSWorld-MCP: even the highest, achieved by Claude 4 Sonnet, is only 33.3 percent, while the lowest, from Qwen2.5-VL-72B-Instruct, is 9.3 percent. This shows that these models still face challenges in invoking tools both correctly and efficiently. For the multi-agent frameworks, Agent-S2.5 achieves an accuracy of 42.1 at the 15-step limit and 49.5 at the 50-step limit, confirming its effectiveness in handling challenging tasks.

**Impact of Tool Invocation on Accuracy and Efficiency.** A comparison with the GUI-only setting shows that, with the exception of Qwen2.5-VL, all six end-to-end models and both agent frameworks achieve higher accuracies and lower ACS after the introduction of MCP tools. Qwen2.5-VL, despite achieving a slight accuracy improvement, shows increased ACS. It indicates poor tool invocation capability and weak decision-making ability, leading to longer average completion times.

Among the remaining models and frameworks, Gemini-2.5-Pro exhibits the most significant improvement. At 50 steps, its overall accuracy rises from 13.3 to 25.7, while ACS decreases from 40.0 to 31.0. Furthermore, a comparison of Tool-Beneficial and Non-Tool-Beneficial Tasks shows that, with the exception of Qwen2.5-VL, all models demonstrate substantial accuracy improvement

Table 1: Performance on OSWorld-MCP.

| Agent Model | Actions | Steps | Tool-Beneficial Tasks | | | Non-Tool-Beneficial Tasks | | | Overall | | |
| | | | Accuracy | TIR | ACS | Accuracy | TIR | ACS | Accuracy | TIR | ACS |
|---|---|---|---|---|---|---|---|---|---|---|---|
| *Open Models* | | | | | | | | | | | |
| Qwen2.5-VL | GUI | 15 | 10.1 | - | 13.0 | 14.4 | - | 12.9 | 11.4 | - | 13.0 |
| | + MCP | 15 | 15.4 $_{5.3\uparrow}$ | 9.5 | 14.1 $_{1.1\uparrow}$ | 12.6 $_{1.8\downarrow}$ | 11.4 | 13.7 $_{0.8\uparrow}$ | 14.5 $_{3.1\uparrow}$ | 10.1 | 14.0 $_{1.0\uparrow}$ |
| | GUI | 50 | 12.3 | - | 30.6 | 17.7 | - | 30.4 | 13.9 | - | 30.5 |
| | + MCP | 50 | 12.0 $_{0.3\downarrow}$ | 3.5 | 39.7 $_{9.1\uparrow}$ | 23.7 $_{6.0\uparrow}$ | 22.5 | 37.4 $_{7.0\uparrow}$ | 15.6 $_{1.7\uparrow}$ | 9.3 | 39.0 $_{8.5\uparrow}$ |
| Qwen3-VL | GUI | 15 | 24.5 | - | 11.5 | 27.3 | - | 11.7 | 25.4 | - | 11.6 |
| | + MCP | 15 | 31.9 $_{7.4\uparrow}$ | 18.5 | 9.6 $_{1.9\downarrow}$ | 34.7 $_{7.4\uparrow}$ | 28.2 | 10.9 $_{0.8\downarrow}$ | 32.8 $_{7.4\uparrow}$ | 21.5 | 10.0 $_{1.6\downarrow}$ |
| | GUI | 50 | 31.9 | - | 25.5 | 38.0 | - | 25.7 | 33.8 | - | 25.6 |
| | + MCP | 50 | 37.3 $_{5.4\uparrow}$ | 22.5 | 17.4 $_{8.1\downarrow}$ | 44.3 $_{6.3\uparrow}$ | 33.9 | 21.2 $_{4.5\downarrow}$ | 39.5 $_{5.7\uparrow}$ | 26.1 | 18.6 $_{7.0\downarrow}$ |
| *Proprietary Models* | | | | | | | | | | | |
| Gemini-2.5-Pro | GUI | 15 | 6.3 | - | 13.6 | 9.8 | - | 14.0 | 7.4 | - | 13.8 |
| | + MCP | 15 | 22.4 $_{16.1\uparrow}$ | 15.2 | 10.5 $_{3.1\downarrow}$ | 6.3 $_{3.5\downarrow}$ | 5.4 | 13.9 $_{0.1\downarrow}$ | 17.4 $_{10.0\uparrow}$ | 12.2 | 11.6 $_{2.2\downarrow}$ |
| | GUI | 50 | 11.5 | - | 39.5 | 17.5 | - | 41.3 | 13.3 | - | 40.0 |
| | + MCP | 50 | 27.6 $_{16.1\uparrow}$ | 16.3 | 26.6 $_{12.9\downarrow}$ | 21.3 $_{3.8\uparrow}$ | 18.0 | 40.7 $_{0.6\downarrow}$ | 25.7 $_{12.4\uparrow}$ | 16.8 | 31.0 $_{9.0\downarrow}$ |
| OpenAI o3 | GUI | 15 | 8.5 | - | 13.8 | 7.8 | - | 14.4 | 8.3 | - | 14.0 |
| | + MCP | 15 | 23.0 $_{14.5\uparrow}$ | 15.2 | 10.8 $_{3.0\downarrow}$ | 5.5 $_{2.3\downarrow}$ | 3.3 | 14.3 $_{0.1\downarrow}$ | 17.6 $_{9.3\uparrow}$ | 11.6 | 11.9 $_{2.1\downarrow}$ |
| | GUI | 50 | 10.7 | - | 43.9 | 17.4 | - | 46.8 | 12.8 | - | 44.8 |
| | + MCP | 50 | 27.0 $_{16.3\uparrow}$ | 17.6 | 27.9 $_{16.0\downarrow}$ | 17.6 $_{0.2\uparrow}$ | 12.3 | 44.4 $_{2.4\downarrow}$ | 24.1 $_{11.3\uparrow}$ | 16.0 | 33.0 $_{11.8\downarrow}$ |
| Seed1.5-VL | GUI | 15 | 27.2 | - | 10.6 | 29.3 | - | 11.5 | 27.9 | - | 10.9 |
| | + MCP | 15 | 31.5 $_{4.3\uparrow}$ | 17.7 | 9.6 $_{1.0\downarrow}$ | 28.9 $_{0.4\downarrow}$ | 28.5 | 11.3 $_{0.2\downarrow}$ | 30.7 $_{2.8\uparrow}$ | 21.0 | 10.1 $_{0.8\downarrow}$ |
| | GUI | 50 | 31.1 | - | 22.5 | 40.2 | - | 26.8 | 33.9 | - | 23.8 |
| | + MCP | 50 | 36.7 $_{5.6\uparrow}$ | 18.3 | 20.1 $_{2.4\downarrow}$ | 41.4 $_{1.2\uparrow}$ | 40.5 | 27.1 $_{0.3\downarrow}$ | 38.2 $_{4.3\uparrow}$ | 25.1 | 22.3 $_{1.5\downarrow}$ |
| Claude-4-Sonnet | GUI | 15 | 29.0 | - | 11.8 | 32.9 | - | 12.0 | 30.2 | - | 11.9 |
| | + MCP | 15 | 37.5 $_{8.5\uparrow}$ | 26.0 | 10.1 $_{1.7\downarrow}$ | 33.0 $_{0.1\uparrow}$ | 30.6 | 11.6 $_{0.4\downarrow}$ | 36.1 $_{5.9\uparrow}$ | 27.4 | 10.5 $_{1.4\downarrow}$ |
| | GUI | 50 | 36.6 | - | 24.7 | 44.0 | - | 25.4 | 38.9 | - | 25.0 |
| | + MCP | 50 | 45.2 $_{8.6\uparrow}$ | 31.5 | 18.9 $_{5.8\downarrow}$ | 44.6 $_{0.6\uparrow}$ | 37.5 | 22.5 $_{2.9\downarrow}$ | 45.0 $_{6.1\uparrow}$ | 33.3 | 20.0 $_{5.0\downarrow}$ |
| *Multi-agent Models* | | | | | | | | | | | |
| Agent-S2.5 | GUI | 15 | 35.8 | - | 11.4 | 38.6 | - | 11.2 | 36.7 | - | 11.3 |
| | + MCP | 15 | 42.9 $_{7.1\uparrow}$ | 28.1 | 9.6 $_{1.8\downarrow}$ | 40.4 $_{1.8\uparrow}$ | 34.2 | 10.9 $_{0.3\downarrow}$ | 42.1 $_{5.4\uparrow}$ | 30.0 | 10.0 $_{1.3\downarrow}$ |
| | GUI | 50 | 46.9 | - | 21.1 | 47.3 | - | 18.4 | 47.1 | - | 20.2 |
| | + MCP | 50 | 49.4 $_{2.5\uparrow}$ | 32.9 | 16.3 $_{4.8\downarrow}$ | 49.6 $_{2.3\uparrow}$ | 40.5 | 18.7 $_{0.3\uparrow}$ | 49.5 $_{2.4\uparrow}$ | 35.3 | 17.0 $_{3.2\downarrow}$ |

in Tool-Beneficial Tasks after the introduction of MCP tools. The largest gain is observed in OpenAI o3 at 50 steps, where accuracy increases from 10.7 to 27.0. In addition, accuracy changes for Non-Tool-Beneficial Tasks are minor. Two possible factors may explain these results: First, Non-Tool-Beneficial Tasks refer only to tasks without efficiency-enhancing tools, yet they may still include tools that, while not improving efficiency, have a positive effect on task completion. Such tools can make it easier for the model to solve the problem, thereby increasing Acc. Second, for tasks that contain no tools beneficial to the task at all, the tools provided can help the model rule out irrelevant solution paths, making it easier to execute the task along the correct path and thus improving task Accuracy. This also explains why ACS often decreases on Non-Tool-Beneficial Tasks. From these results, we derive the following conclusion:

> **Finding 1:** *MCP tools significantly enhance LMMs' performance in computer-use, improving accuracy and reducing completion steps for most models. The effectiveness varies across different LMMs, indicating disparities in tool utilization capabilities.*

**Impact of Tool Invocation Rate on Accuracy and Average Completion Steps.** Our experiments reveal that, across different models, the tool invocation rate (TIR) and accuracy (Acc) generally exhibit a positive correlation, whereas Average Completion Steps (ACS) show no obvious correlation with TIR. We computed TIR, Acc, and ACS for each model under varying maximum step limits, across Tool-Beneficial Tasks, Non-Tool-Beneficial Tasks, and the entire task set. These results were aggregated into a single chart (Figure 5.(a)). As shown in the figure, for a given model, higher TIR values tend to correspond to higher accuracies, indicating a clear positive relationship between tool invocation and task success. Notably, this relationship remains stable regardless of differences in step limits or task sets. This strongly supports the soundness of our MCP tool design. In contrast, ACS does not show a clear correlation with TIR. Further analysis of ACS–TIR patterns across different task sets and step limits suggests that, under the same settings, an increase in TIR can sometimes coincide with a decrease in ACS. We hypothesize two possible reasons for this phenomenon: a). TIR reflects the proportion of correct tool invocations. A higher TIR indicates a higher proportion of correct tool usage, which can enable the model to complete tasks more efficiently. b). The complexity of OSWorld-MCP tasks varies significantly, and the overall task completion rate remains relatively

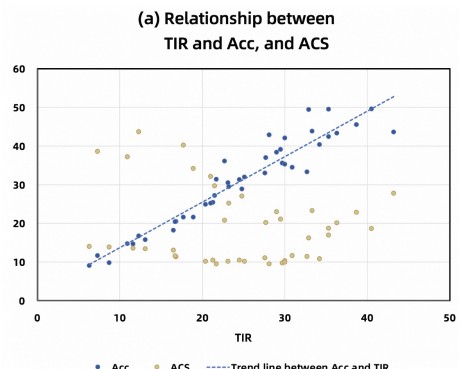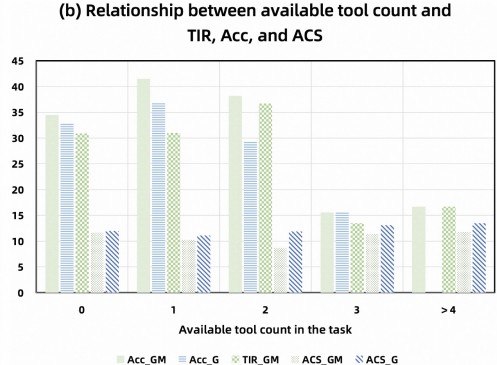

Figure 5: (a): The relationships between TIR, Acc, and ACS. (b): The performance of Claude-4-Sonnet across task sets with different numbers of available tools. GM: GUI + MCP , G: GUI Only.

Table 2: Performance of Gemini-2.5-Pro on OSWorld-MCP under different configurations.

| Agent Model | Settings | Tool-Beneficial Tasks | | | Non-Tool-Beneficial Tasks | | | Overall | | |
|---|---|---|---|---|---|---|---|---|---|---|
| | | Accuracy | TIR | ACS | Accuracy | TIR | ACS | Accuracy | TIR | ACS |
| Gemini-2.5-Pro | Base (MCP) | 22.5 | 15.2 | 10.3 | 7.2 | 6.3 | 14.1 | 17.8 | 12.5 | 11.5 |
| | w/ Tools Shuffle | $22.3_{0.2\downarrow}$ | $14.8_{0.4\downarrow}$ | $10.4_{0.1\uparrow}$ | $13.1_{5.9\uparrow}$ | $12.6_{6.3\uparrow}$ | $13.9_{0.2\downarrow}$ | $19.5_{1.7\uparrow}$ | $14.1_{1.6\uparrow}$ | $11.5_{0.0-}$ |
| | w/o Tools RAG | $16.3_{6.2\downarrow}$ | $11.2_{4.0\downarrow}$ | $8.7_{1.6\downarrow}$ | $7.4_{0.2\uparrow}$ | $7.2_{0.9\uparrow}$ | $9.3_{4.8\downarrow}$ | $13.6_{4.2\downarrow}$ | $10.1_{2.4\downarrow}$ | $8.9_{2.6\downarrow}$ |

low. When the maximum step limit is raised, models tend to make more attempts in solving complex tasks, which can counteract or obscure the efficiency gains that come from correct tool usage.

> **Finding 2:** *Tool Invocation Rate (TIR) positively correlates with task accuracy, but its relationship with ACS is complex and non-linear, suggesting that the impact of tool use on efficiency depends on various factors including task difficulty and model-specific strategies.*

**Impact of the number of available tools.** As shown in Figure 5.(b), we conducted experiments using the best-performing model from the previous evaluations, Claude 4 Sonnet, on the manually annotated task set described in Section 3.3. The tasks were grouped according to the number of available tools. For each configuration, we computed Acc, TIR, and ACS.

Firstly, in the results for the GUI-only configuration, we observe that as the number of available tools increases, Acc tends to decrease and ACS tends to increase. This indicates that task difficulty rises with the number of available tools. Secondly, in the results for the GUI plus MCP Tools configuration, we found that when the number of available tools was relatively small, Acc increased and ACS decreased. However, as the number of available tools grew, both Acc and TIR dropped sharply, and ACS gradually rose. A possible explanation is that with fewer available tools, the tasks are relatively easier, and the model is more likely to select the tools that are useful for the task, thereby completing it with higher accuracy and efficiency. When the number of available tools increases, the tasks become more complex. Even though more efficient tools might be available, these tasks often require multiple-tool combinations, making it difficult for the model to accurately select the most relevant tools, leading to reduced task accuracy and higher average completion steps. Thirdly, when comparing the GUI-only configuration with the GUI plus MCP Tools configuration, the latter consistently achieved lower ACS and higher accuracy overall. However, in cases where ACS was similar in both configurations, we found that the GUI plus MCP Tools configuration sometimes resulted in lower accuracy. We suspect that the use of MCP tool combinations is more challenging for LMMs than combining GUI operations.

> **Finding 3:** *MCP tools generally improve performance in complex tasks, but their efficacy diminishes in extremely complex scenarios requiring tool combinations. This indicates that combining multiple tools is more challenging than combining GUI operations.*

### 4.3 ABLATION STUDY

To more comprehensively evaluate the tool invocation and decision-making capabilities of LMMs, we conducted a series of ablation studies. We selected Gemini-2.5-Pro, the model with the largest accuracy gain from MCP tools over the GUI-only setting, as the test subject for these experiments.

**Impact of the Number of Callable Tools on Model Accuracy.** In the default OSWorld-MCP setup, the set of tools available at each step is a subset obtained via Retrieval-Augmented Generation (RAG), filtered to match the current application in use. To investigate the impact of the number of callable tools on model performance, we removed the RAG filtering and allowed the model to choose freely from all 158 tools for each task. As shown in Table 2, removing RAG led to a noticeable performance drop: the overall Acc decreased from 17.8 to 13.6. We attribute this to the fact that descriptions of all 158 tools result in excessively long tool contexts, which markedly reduce the model's tendency to use tools, thereby impairing accurate tool invocation. For Tool-beneficial Tasks, removing RAG reduced Acc from 22.5 to 16.3 and TIR from 15.2 to 11.2, indicating a pronounced decline in the model's inclination to invoke tools for problem solving. In contrast, for Non-tool-beneficial Tasks, although Acc remained virtually unchanged after removing RAG, TIR increased from 6.3 to 7.2, suggesting that the model became more inclined to employ the GUI rather than tools to solve the tasks.

**Impact of Tool Description Order in Prompts.** In the default OSWorld-MCP configuration, tool descriptions are provided to the LMM in alphabetical order. To investigate the impact of tool description ordering on model performance, we randomly shuffled the descriptions prior to passing them to the LMM and re-evaluated on OSWorld-MCP. As shown in Table 2, random ordering increased overall Acc from 17.8 to 19.5. This result indicates that the ordering of tool descriptions in the prompt has a substantial effect on model performance in OSWorld-MCP. While the model's performance on Tool-beneficial Tasks was nearly unchanged before and after shuffling, differences were considerable for Non-tool-beneficial Tasks. This may be because, with fewer tools, the model tends to invoke the corresponding tool when one is available, whereas in the absence of available tools, the description order may implicitly suggest alternative solution strategies. For consistency in evaluation, tool descriptions are presented to the LMM in lexicographical order in OSWorld-MCP.

### 4.4 CASE STUDY

In order to demonstrate the tool invocation capability of the GUI Agent, we present a complex example in LibreOffice Calc: *Copy the "Revenue" column along with the header to a new sheet named "Sheet2"*. This task requires creating a new sheet and copying the specified column. The end-to-end Gemini-2.5-Pro agent accomplishes this by utilizing tools for creating a sheet and copying data, then switching to Sheet2 to verify the operation, as illustrated in Figure 9. In contrast, the agent without MCP tools fails to select the specific column, demonstrating how tools can serve as a valuable complement to the agent's capabilities. More cases are analyzed in Appendix A.1.

## 5 CONCLUSION

We introduce OSWorld-MCP, a fair and comprehensive benchmark for evaluating Large Multimodal Models (LMMs) in computer-use scenarios by jointly assessing graphical user interface (GUI) operation skills and MCP tool-invocation capabilities. Using an automated pipeline combined with meticulous manual validation, we construct a high-quality and diverse collection of MCP tools that supports realistic and balanced evaluation within the OSWorld framework. Experiments on seven state-of-the-art LMMs and multi-agent frameworks demonstrate that tool invocation can substantially improve robustness and efficiency, while also revealing trade-offs between usage frequency and overall performance. Looking ahead, extending OSWorld-MCP to more complex, dynamic, and collaborative environments, as well as incorporating human-centred evaluation metrics, will further advance the development of general-purpose, efficient, and trustworthy computer-use agents.

## ETHICS STATEMENT

The OSWorld-MCP dataset is constructed entirely from publicly available software environments and tasks in OSWorld. All MCP tools included in the benchmark are either automatically generated and manually validated by the authors or selected from existing open-source MCP servers, ensuring that no proprietary, confidential, or personally identifiable information is included. The dataset contains only synthetic interaction records between multimodal agents and computer application environments; no human subject data are collected. We release OSWorld-MCP solely for research and educational purposes to advance the evaluation of multimodal agents in realistic computer-use scenarios. Researchers using this dataset should comply with all applicable laws, institutional guidelines, and license terms. The authors bear responsibility for ensuring that the dataset is free of harmful or unethical content and that its use will not compromise privacy or security.

## REPRODUCIBILITY STATEMENT

We will release all resources necessary to reproduce our work, including the OSWorld-MCP dataset, the complete set of 158 validated MCP tools, and the automated code-generation pipeline used to create them. Detailed documentation will be provided to ensure that researchers can replicate our experiments under the same conditions, including task definitions, maximum step settings, and all metric implementations (Accuracy, Tool Invocation Rate, and Average Completion Steps). All resources will be hosted in an open-access repository upon publication.

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

# A APPENDIX

## A.1 MORE CASES

We present and analyze additional tool call examples to demonstrate how tool calls enable agents to complete diverse and complex tasks more efficiently and accurately.

### A.1.1 CASE: VSCODE SETTINGS

The goal of the task is "Please help me modify VS Code setting to hide all '__pycache__' folders in the explorer view". Executed actions are as follows:

1. `osworld_mcp_code.add_files_exclude`
   - `pattern="**/__pycache__"`
2. `computer_use`
   - `action="left_click"`
   - `x=1131.0`
   - `y=712.5`
3. `computer_use`
   - `action="terminate"`
   - `status="success"`

Figure 6 showcases the executing process. According to the evaluation program, the specific folder is hidden in the explorer view.

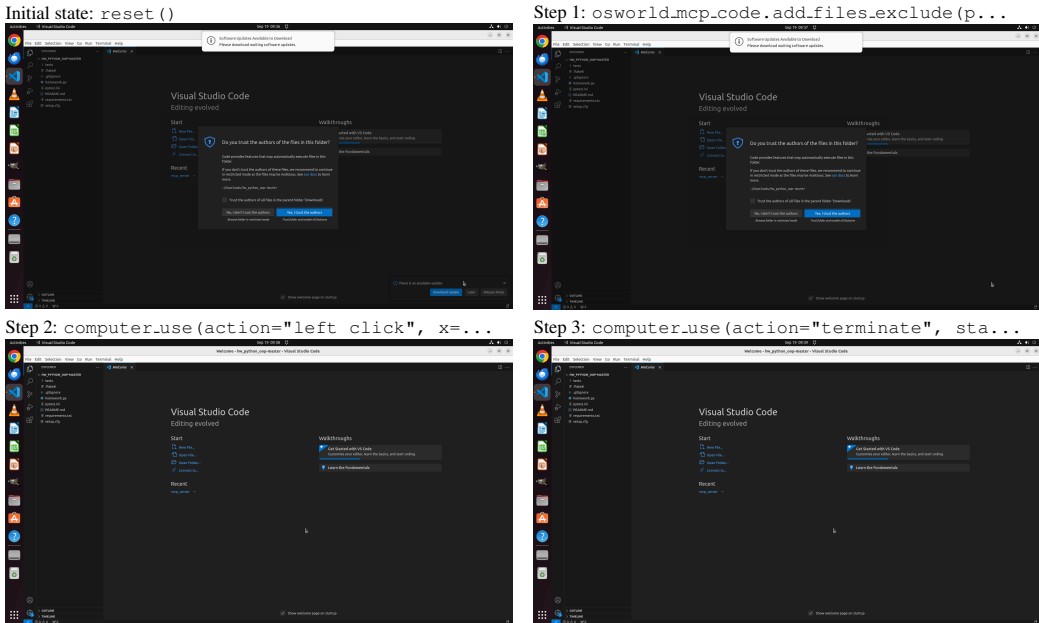

Figure 6: Please help me modify VS Code setting to hide all "__pycache__" folders in the explorer view.

### A.1.2 CASE: FILESYSTEM OPERATION

The goal of the task is "I have an image of my receipt located in /home/user. I'm looking to transform it into a PDF file. Can you assist me with this task? Save the resulting PDF as 'receipt.pdf' on the desktop". Executed actions are as follows:

1. `osworld_mcp_os.convert_image_format`
   - `image_path="/home/user/receipt.jpg"`
   - `output_format="pdf"`
   - `output_path="/home/user/Desktop/receipt.pdf"`
2. `filesystem_list_directory`
   - `path="/home/user"`
3. `osworld_mcp_os.convert_image_format`
   - `image_path="/home/user/OIP.jpg"`
   - `output_format="pdf"`
   - `output_path="/home/user/Desktop/receipt.pdf"`
4. `computer_use`
   - `action="terminate"`
   - `status="success"`

The agent first utilized the tool to convert an image to PDF format, but the tool response indicated the specified file or directory did not exist. By listing the files in the home directory path, the agent obtained the correct filename. The subsequent conversion attempt successfully produced the desired PDF file from the original image, as demonstrated in Figure 7.

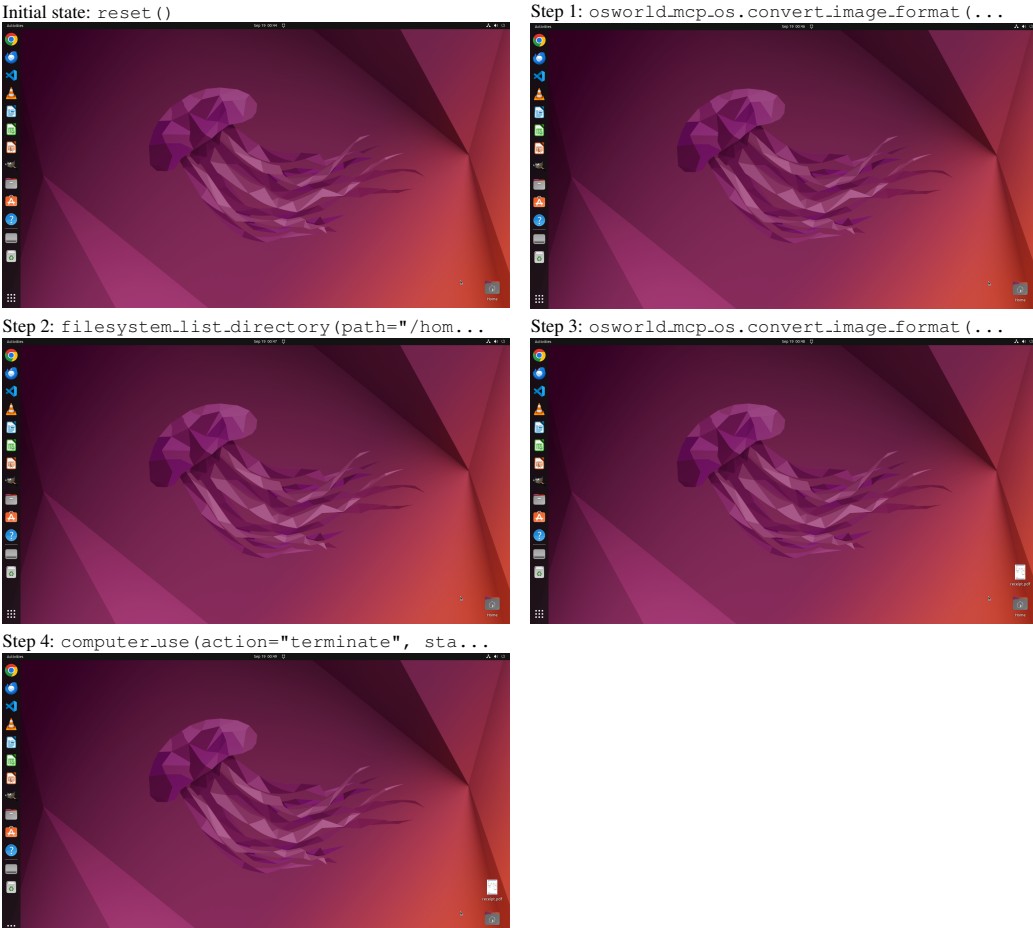

Figure 7: I have an image of my receipt located in /home/user. I'm looking to transform it into a PDF file. Can you assist me with this task? Save the resulting PDF as "receipt.pdf" on the desktop.

### A.1.3 CASE: SLIDE DUPLICATION

The goal of the task is "Please duplicate the last two slides and insert the copies in alternating order, so the sequence becomes: original slide A, original slide B, then duplicated slide A, duplicated slide B". Executed actions are as follows:

1. `osworld_mcp_libreoffice_impress.get_slide_count`
2. `osworld_mcp_libreoffice_impress.duplicate_slide`
   - `slide_index=24`
   - `target_position=25`
3. `osworld_mcp_libreoffice_impress.get_slide_count`
4. `osworld_mcp_libreoffice_impress.duplicate_slide`
   - `slide_index=24`
   - `target_position=25`
5. `osworld_mcp_libreoffice_impress.duplicate_slide`
   - `slide_index=23`
   - `target_position=24`
6. `osworld_mcp_libreoffice_impress.duplicate_slide`
   - `slide_index=24`
   - `target_position=25`
7. `osworld_mcp_libreoffice_impress.save`
8. `computer_use`
   - `action="terminate"`
   - `status="success"`

The agent initially determined the positions of the last two slides (indexed 23 and 24) through tool calls and attempted to directly duplicate the slides. However, the operation failed at steps 2 and 4 due to the incorrect parameter of target position. After several attempts, the agent successfully duplicated the final two slides to positions 25 at step 5, and 26 at step 6. As shown in Figure 8, the agent ultimately completing the task.

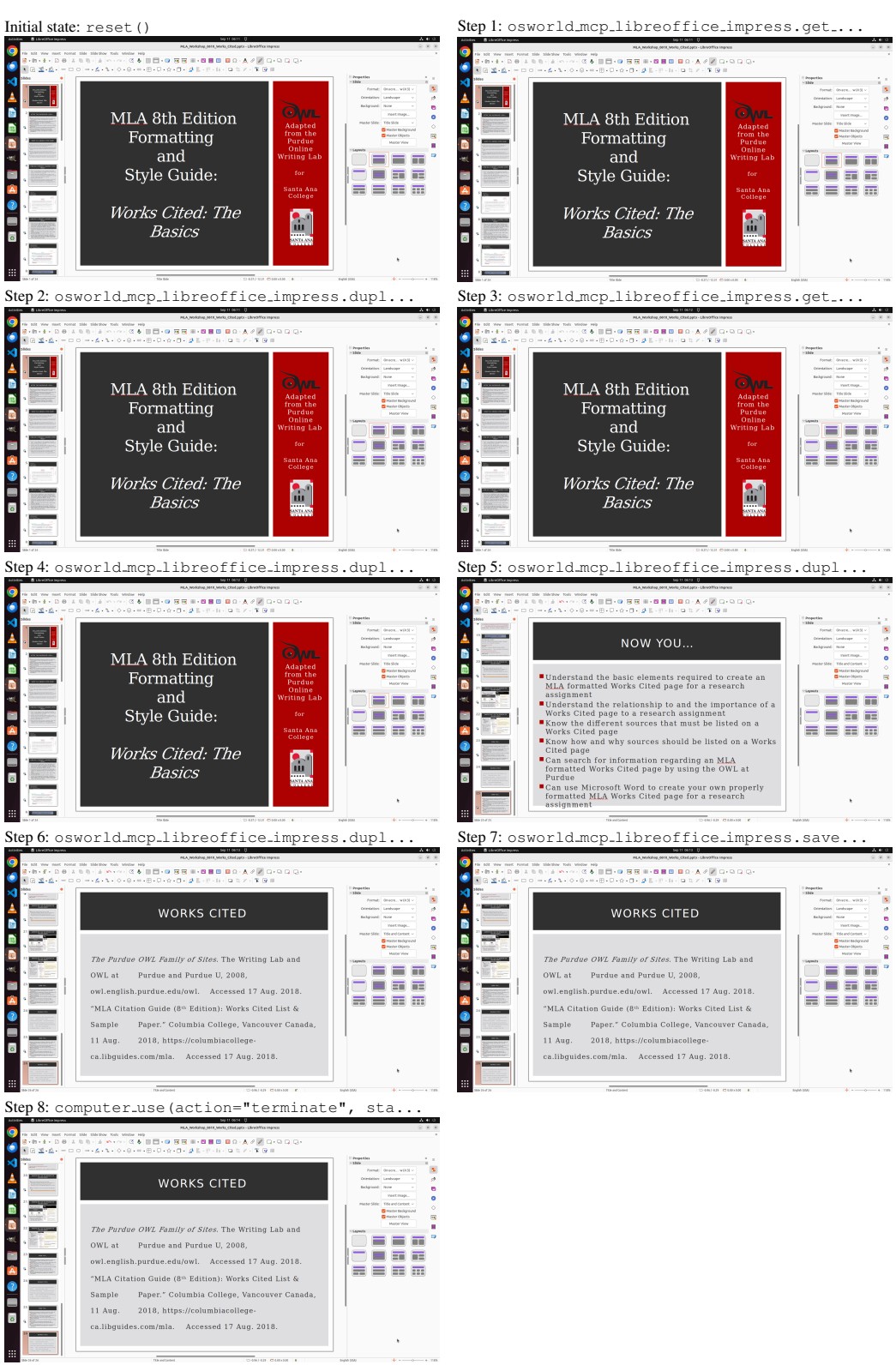

Figure 8: Please duplicate the last two slides and insert the copies in alternating order, so the sequence becomes: original slide A, original slide B, then duplicated slide A, duplicated slide B.

### A.1.4 CASE: SPREADSHEET COPYING

The goal of the task is "Copy the 'Revenue' column along with the header to a new sheet named 'Sheet2' ". Executed actions are as follows:

1. `osworld_mcp_libreoffice_calc.copy_cells_between_sheets`
   - `source_range="A1:A20"`
   - `target_sheet_name="Sheet2"`
   - `target_start_cell="A1"`
2. `osworld_mcp_libreoffice_calc.copy_cells_between_sheets`
   - `source_range="A1:A20"`
   - `target_sheet_name="Sheet2"`
   - `target_start_cell="A1"`
3. `osworld_mcp_libreoffice_calc.get_workbook_info`
4. `osworld_mcp_libreoffice_calc.copy_cells_between_sheets`
   - `source_range="A1:A20"`
   - `target_sheet_name="Sheet2"`
   - `target_start_cell="A1"`
5. `osworld_mcp_libreoffice_calc.switch_active_sheet`
   - `sheet_name="Sheet2"`
6. `osworld_mcp_libreoffice_calc.switch_active_sheet`
   - `sheet_name="Sheet1"`
7. `osworld_mcp_libreoffice_calc.copy_cells_between_sheets`
   - `source_range="A1:A20"`
   - `target_sheet_name="Sheet2"`
   - `target_start_cell="A1"`
8. `osworld_mcp_libreoffice_calc.switch_active_sheet`
   - `sheet_name="Sheet2"`
9. `computer_use`
   - `action="terminate"`
   - `status="success"`

According to Figure 9, the "Revenue" column has been copied to "Sheet2" successfully.

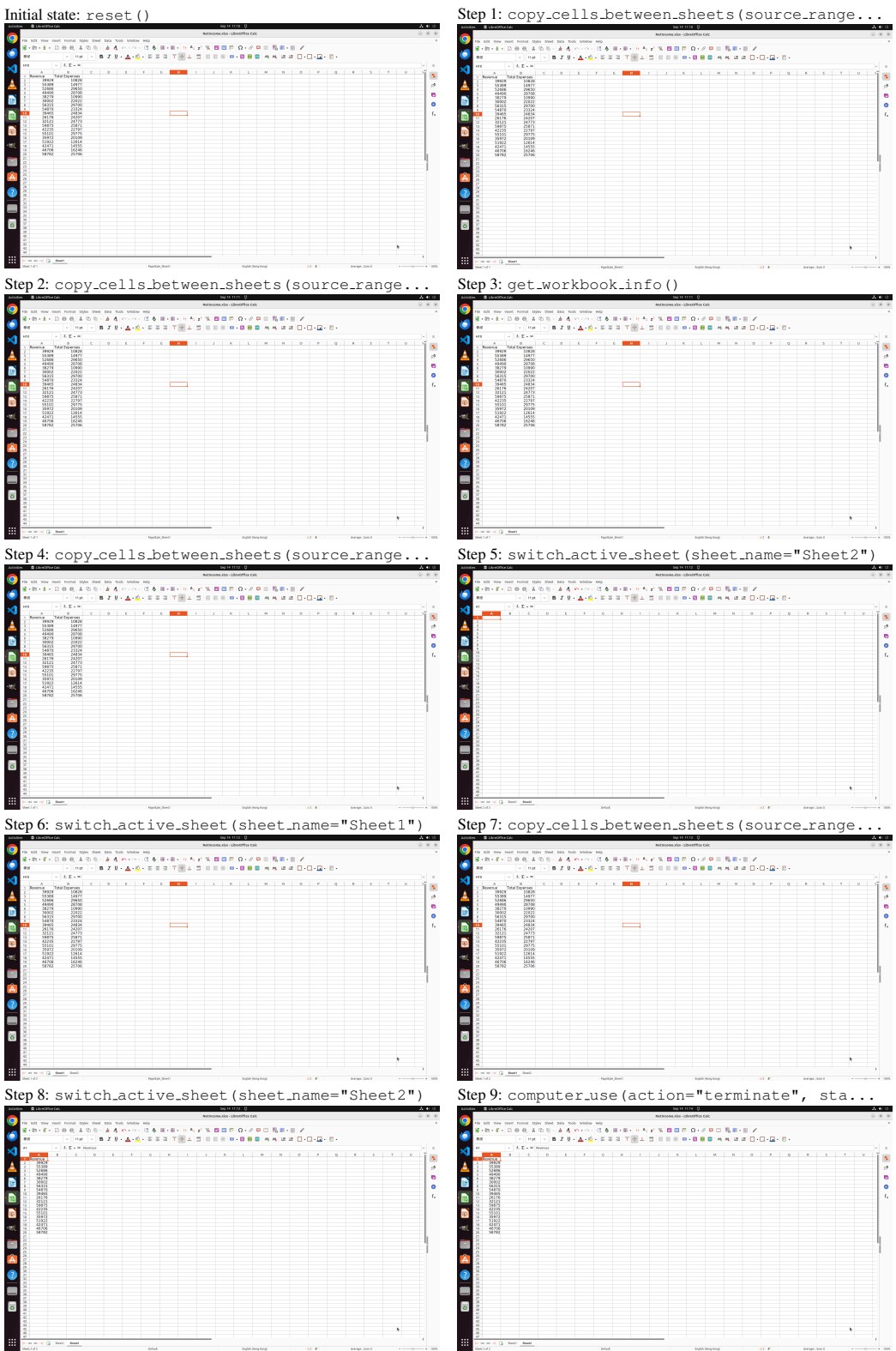

Figure 9: Copy the "Revenue" column along with the header to a new sheet named "Sheet2".

## A.2 BAD CASES

We present and analyze additional tool call examples to demonstrate how tool calls enable agents to complete diverse and complex tasks more efficiently and accurately.

### A.2.1 CASE: REASONING FAILURE

The goal of the task is "Lately I have changed my English name to Thomas. I want to update my username. Could you help me change the username in chrome profiles to Thomas?". The Claude-4-Sonnet model successfully open the profile setting in Figure 10, filling the text box with new name Thomas. However, it immediately report finishing task without clicking anywhere else or press enter, leaving the name unchanged, finally caused the task failed.

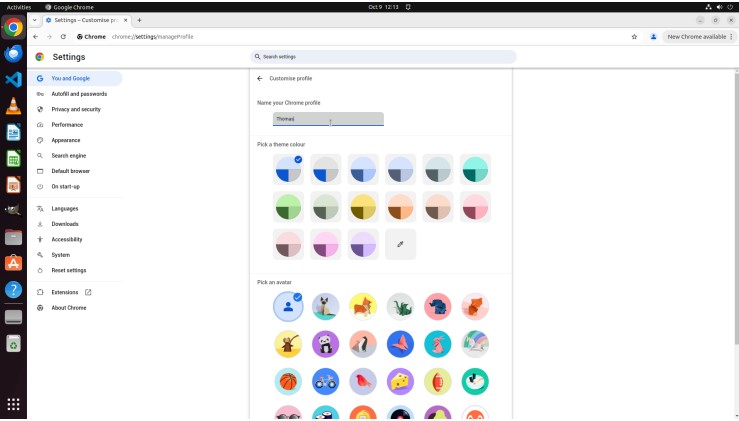

Figure 10: Reasoning Failure

In another task, the goal is "Check the names in column "Names with duplicates" and put the unique ones in column "Unique Names". Keep the original order of the first occurrences", as shown in Figure 11. The task requires multiple read-and-write operation, which can be efficiently and accurately addressed by tool `osworld_mcp_libreoffice_calc.set_cell_value` in one step. Instead of utilizing MCP tools, Qwen2.5-VL attempted to call GUI operation only, clicking a wrong cell during processing, which finally lead to an inaccurate input position and failed to finish the task. The resuls is demonstrated in Figure 12.

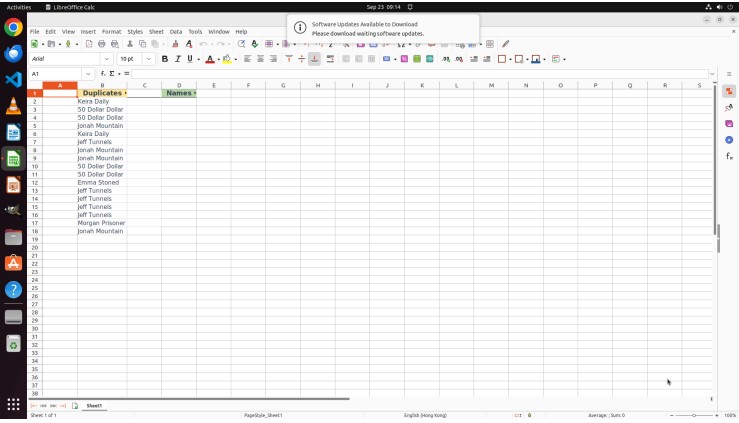

Figure 11: Reasoning Failure (before)

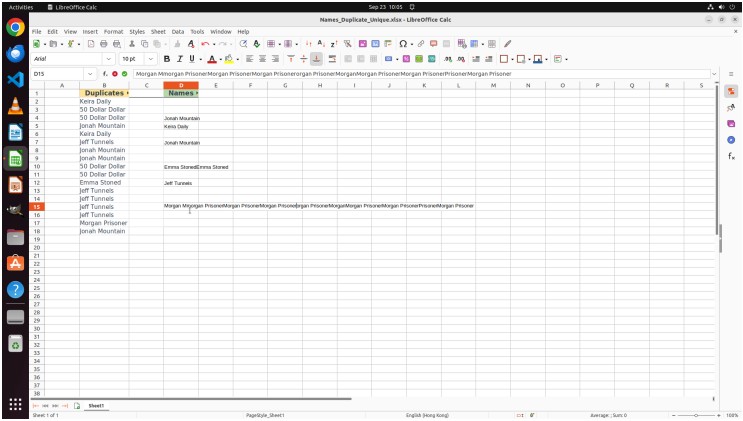

Figure 12: Reasoning Failure (after)

### A.2.2 CASE: GROUNDING FAILURE

The goal of the task is "Can you make a new folder for me on the bookmarks bar in my internet browser? Let's call it 'Favorites". For OpenAI o3 model, it first opened the bookmark page by `osworld_mcp_google_chrome.open_bookmarks`, and attempted to right-click the bookmark bar for opening the context-menu, as shown in 13. However, the model clicked the wrong position and failed to open the context-menu for times, leading to a failure of this task in 13.

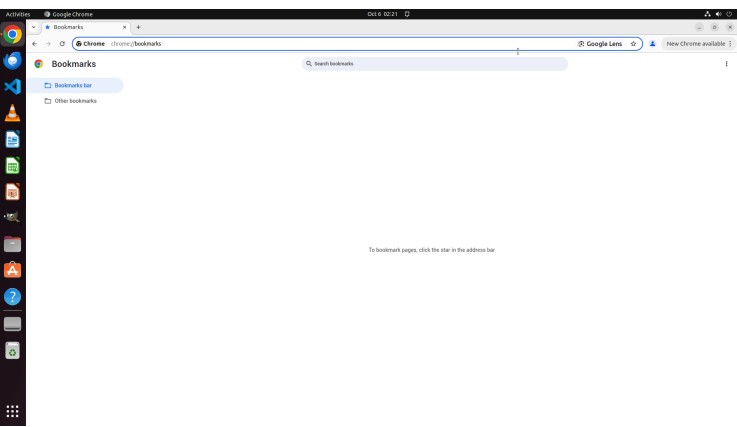

Figure 13: Grounding Failure (before)

### A.2.3 CASE: MCP TOOL INVOCATION ERROR

The goal of the task is "On my surface pro whenever I launch Chrome it always opens ̈funbrain.com.Ï don't want this. I cleared my cache but it still happens. What should I do?". For Qwen2.5-VL model, the thought of first steps in 15 are:

```
To address the issue of Chrome always opening 'funbrain.com,' I
↪   need to check if there are any extensions that might be
↪   causing this behavior. Opening the extensions page will allow
↪   me to review and disable any suspicious or unwanted
↪   extensions.\nAction: Click on the three-dot menu in the top
↪   right corner of Chrome and select 'More tools' then
↪   'Extensions.'
```

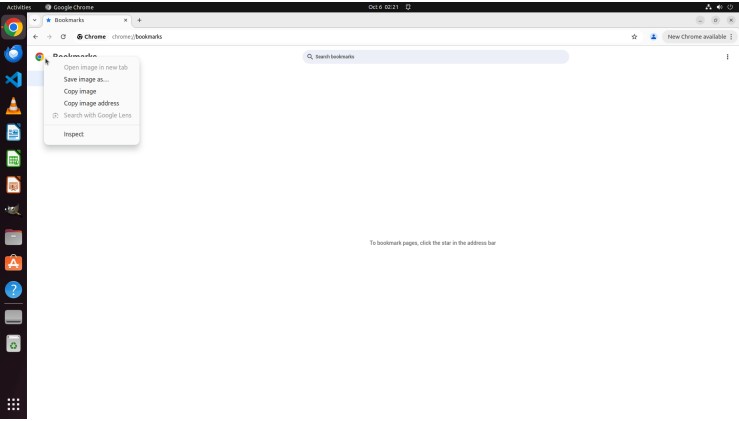

Figure 14: Grounding Failure (after)

However, the following made action is `osworld_mcp_google_chrome.open_extensions`, which misaligned with the previous thought. The mismatching between thought and action may cause potential impact.

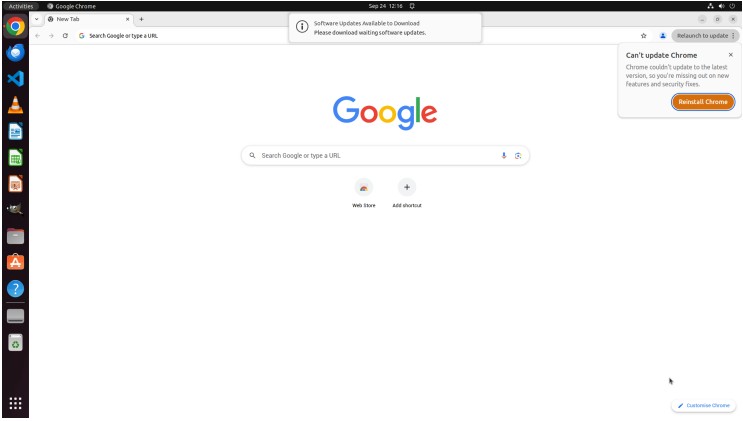

Figure 15: MCP Tool Invocation Error

### A.2.4 CASE: MCP TOOL PARAMETER ERROR

The goal of the task is "So, I've been dabbling with coding a Snake game in Python, and I finally got it up and running. It's pretty cool, but it's not without its quirks. The biggest issue I'm facing right now is that the snake can't seem to eat the food, no matter what. Could you help me tweak the code so the snake can actually eat the food? Thanks a bunch!". For Qwen2.5-VL model, it firstly checked the code with thought in 16:

```
To ensure the snake can eat the food, I need to check the logic in
↪  `main.py` where the collision between the snake and the food
↪  is detected. The current issue might be related to how
↪  positions are compared or updated.\nAction: Click on the
↪  'main.py' file in the explorer to review and modify the
↪  collision detection logic between the snake and the food.
```

However, it raised a tool calling

```
{"action_type": "osworld_mcp_code", "parameters": {"file_path":
↪  "/path/to/main.py", "line": 26, "character": 5}
```

which passed a totally wrong tool name and parameter to the MCP server, leading to a failure of the task.

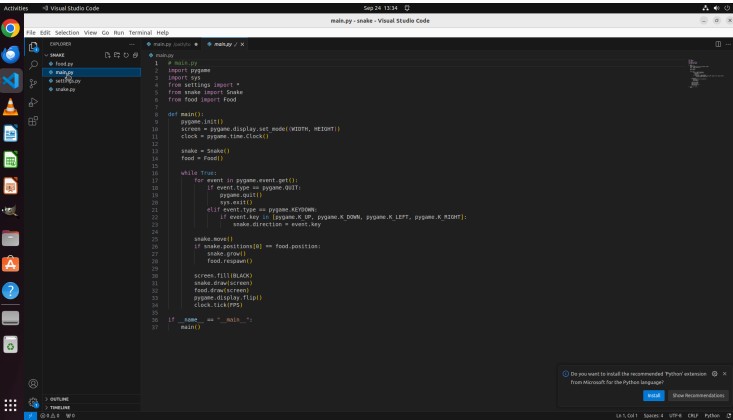

Figure 16: MCP Tool Parameter error

Table 3: Performance on OSWorld-MCP.

| Agent | Steps | Method | Accuracy (mean±std) | Accuracy (CI) | TIR (mean±std) | TIR (CI) | ACS (mean±std) | ACS (CI) |
|---|---|---|---|---|---|---|---|---|
| Agent S2.5 | 15 | GUI Only | 36.7 ± 2.7 | (33.6, 39.7) | - | - | 11.3 ± 0.1 | (11.1, 11.5) |
| Agent S2.5 | 15 | GUI+MCP | 42.1 ± 1.5 | (40.4, 43.9) | 30.0 ± 1.4 | (28.4, 31.6) | 10.0 ± 0.2 | (9.8, 10.2) |
| Agent S2.5 | 50 | GUI Only | 47.1 ± 2.7 | (44.0, 50.2) | - | - | 20.2 ± 0.5 | (19.6, 20.8) |
| Agent S2.5 | 50 | GUI+MCP | 49.5 ± 2.1 | (47.2, 51.8) | 35.3 ± 0.6 | (34.6, 35.9) | 17.0 ± 0.5 | (16.4, 17.6) |
| Seed-vl | 15 | GUI Only | 27.9 ± 1.3 | (26.4, 29.4) | - | - | 10.9 ± 0.0 | (10.8, 10.9) |
| Seed-vl | 15 | GUI+MCP | 30.7 ± 1.0 | (29.6, 31.8) | 21.0 ± 1.0 | (19.9, 22.2) | 10.1 ± 0.0 | (10.1, 10.2) |
| Seed-vl | 50 | GUI Only | 33.9 ± 1.7 | (32.0, 35.8) | - | - | 23.8 ± 0.2 | (23.6, 24.0) |
| Seed-vl | 50 | GUI+MCP | 38.2 ± 1.5 | (36.5, 39.8) | 25.1 ± 1.3 | (23.6, 26.6) | 22.3 ± 0.3 | (21.9, 22.6) |
| Qwen3-VL | 15 | GUI Only | 25.4 ± 1.3 | (23.8, 26.9) | - | - | 11.6 ± 0.1 | (11.5, 11.7) |
| Qwen3-VL | 15 | GUI+MCP | 32.8 ± 0.2 | (32.6, 33.1) | 21.5 ± 1.5 | (19.8, 23.2) | 10.0 ± 0.2 | (9.8, 10.2) |
| Qwen3-VL | 50 | GUI Only | 33.8 ± 2.0 | (31.5, 36.0) | - | - | 25.6 ± 0.4 | (25.1, 26.1) |
| Qwen3-VL | 50 | GUI+MCP | 39.5 ± 0.1 | (39.3, 39.6) | 26.1 ± 2.0 | (23.8, 28.3) | 18.6 ± 0.4 | (18.1, 19.0) |
| Claude 4 Sonnet | 15 | GUI Only | 30.2 ± 1.7 | (28.3, 32.1) | - | - | 11.9 ± 0.1 | (11.8, 12.0) |
| Claude 4 Sonnet | 15 | GUI+MCP | 36.1 ± 0.3 | (35.7, 36.5) | 27.4 ± 0.0 | (27.4, 27.4) | 10.5 ± 0.2 | (10.3, 10.7) |
| Claude 4 Sonnet | 50 | GUI Only | 38.9 ± 1.6 | (37.1, 40.7) | - | - | 25.0 ± 0.2 | (24.8, 25.2) |
| Claude 4 Sonnet | 50 | GUI+MCP | 45.0 ± 0.5 | (44.4, 45.6) | 33.3 ± 0.1 | (33.1, 33.5) | 20.0 ± 0.5 | (19.4, 20.6) |
| Gemini-2.5 Pro | 15 | GUI Only | 7.4 ± 3.1 | (3.9, 10.9) | - | - | 13.8 ± 0.5 | (13.3, 14.3) |
| Gemini-2.5 Pro | 15 | GUI+MCP | 17.4 ± 1.4 | (15.9, 19.0) | 12.2 ± 0.6 | (11.6, 12.8) | 11.6 ± 0.1 | (11.5, 11.7) |
| Gemini-2.5 Pro | 50 | GUI Only | 13.3 ± 1.8 | (11.3, 15.3) | - | - | 40.0 ± 1.7 | (38.1, 41.9) |
| Gemini-2.5 Pro | 50 | GUI+MCP | 25.7 ± 1.2 | (24.4, 27.0) | 16.8 ± 1.3 | (15.3, 18.3) | 31.0 ± 0.2 | (30.7, 31.2) |
| OpenAI o3 | 15 | GUI Only | 8.3 ± 3.8 | (4.1, 12.6) | - | - | 14.0 ± 0.2 | (13.8, 14.2) |
| OpenAI o3 | 15 | GUI+MCP | 17.6 ± 0.7 | (16.8, 18.4) | 11.6 ± 0.5 | (11.0, 12.1) | 11.9 ± 0.1 | (11.8, 12.0) |
| OpenAI o3 | 50 | GUI Only | 12.8 ± 0.4 | (12.3, 13.2) | - | - | 44.8 ± 0.4 | (44.4, 45.2) |
| OpenAI o3 | 50 | GUI+MCP | 24.1 ± 0.9 | (23.1, 25.2) | 16.0 ± 0.6 | (15.3, 16.6) | 33.0 ± 0.7 | (32.2, 33.8) |
| Qwen2.5-VL | 15 | GUI Only | 11.4 ± 1.8 | (9.4, 13.5) | - | - | 13.0 ± 0.1 | (12.8, 13.1) |
| Qwen2.5-VL | 15 | GUI+MCP | 14.5 ± 0.2 | (14.3, 14.8) | 10.1 ± 0.6 | (9.4, 10.7) | 14.0 ± 0.0 | (13.9, 14.0) |
| Qwen2.5-VL | 50 | GUI Only | 13.9 ± 0.6 | (13.3, 14.6) | - | - | 30.5 ± 0.3 | (30.2, 30.9) |
| Qwen2.5-VL | 50 | GUI+MCP | 15.6 ± 0.4 | (15.2, 16.0) | 9.3 ± 0.6 | (8.7, 10.0) | 39.0 ± 0.5 | (38.4, 39.6) |

## A.3 VARIANCE MEASURES AND CONFIDENCE INTERVALS

To further accurately evaluate the performance of each agent, we conducted three independent runs for each agent. Table 3 below reports the mean, standard deviation, and 95% confidence interval of Accuracy, TIR, and ACS for each model under different settings.

The experimental results show that in most cases, after introducing MCP, the performance differences between runs under the same settings become smaller, and the models demonstrate greater stability. In certain cases, this may be related to the relatively low Tool Invocation Rate of the model.

## A.4 MCP TOOL LIST

We present the complete list of 158 MCP tools as follows.

Table 4: Complete List of MCP Tools

| No. | MCP Server | Domain | MCP Tool |
|---|---|---|---|
| 1 | osworld_mcp | VS Code | add_folder 
 Adds a folder to the last active window in VS-Code |
| 2 | osworld_mcp | VS Code | compare_files 
 Compares two files in VSCode |
| 3 | osworld_mcp | VS Code | disable_extension 
 Disables a specific extension for the next instance of VSCode |
| 4 | osworld_mcp | VS Code | goto_file 
 Opens a file at a specific line and character position |
| 5 | osworld_mcp | VS Code | install_extension 
 Installs an extension or updates it in VSCode |
| 6 | osworld_mcp | VS Code | launch_vscode 
 Launches Visual Studio Code with the specified file path or directory |
| 7 | osworld_mcp | VS Code | list_extensions 
 Lists installed extensions in VSCode |

| No. | MCP Server | Domain | MCP Tool |
|---|---|---|---|
| 8 | osworld_mcp | VS Code | `remove_folder`
Removes a folder from the last active window in VSCode |
| 9 | osworld_mcp | VS Code | `toggle_sync`
Toggles synchronization on or off in VSCode |
| 10 | osworld_mcp | VS Code | `uninstall_extension`
Uninstalls an extension from VSCode |
| 11 | osworld_mcp | VS Code | `update_extensions`
Updates all installed extensions in VSCode to the latest version |
| 12 | osworld_mcp | VS Code | `add_files_exclude`
Add a glob pattern to the Files: Exclude setting |
| 13 | osworld_mcp | VS Code | `replace_text`
Open VSCode search and replace panel, input search and replacement text, and execute replacement |
| 14 | osworld_mcp | VS Code | `search_text`
Open VSCode search panel, input the search term, and execute search |
| 15 | osworld_mcp | VS Code | `set_auto_save_delay`
Set the auto save delay in milliseconds |
| 16 | osworld_mcp | VS Code | `set_color_theme`
Change the editor color theme |
| 17 | osworld_mcp | VS Code | `set_focus_editor_on_break`
Set the Debug: Focus Editor On Break setting to true or false |
| 18 | osworld_mcp | VS Code | `set_python_diagnostics_override`
Override severity for a specific Python analysis diagnostic rule |
| 19 | osworld_mcp | VS Code | `set_word_wrap_column`
Set the number of columns at which the editor will word wrap |
| 20 | osworld_mcp | VS Code | `set_wrap_tabs`
Enable or disable wrap tabs in the editor |
| 21 | osworld_mcp | Google Chrome | `bookmark_page`
Bookmarks the current page in the browser (equivalent to Ctrl+D) |
| 22 | osworld_mcp | Google Chrome | `bring_back_last_tab`
Restores the last-closed tab in the browser (equivalent to Ctrl+Shift+T) |
| 23 | osworld_mcp | Google Chrome | `chrome_open_tabs_setup`
Opens the entered URL |
| 24 | osworld_mcp | Google Chrome | `delete_browsing_data`
Opens the 'Clear browsing data' dialog in the browser (equivalent to Ctrl+Shift+Del) |
| 25 | osworld_mcp | Google Chrome | `open_appearance_settings`
Opens the appearance settings page in the browser |
| 26 | osworld_mcp | Google Chrome | `open_bookmarks`
Opens the bookmarks page in the browser |
| 27 | osworld_mcp | Google Chrome | `open_extensions`
Opens the extensions management page in the browser |
| 28 | osworld_mcp | Google Chrome | `open_password_settings`
Opens the password/autofill settings page in the browser |

| No. | MCP Server | Domain | MCP Tool |
|-----|-----------|--------|----------|
| 29 | osworld_mcp | Google Chrome | `open_privacy_settings`
Opens the privacy settings page in the browser |
| 30 | osworld_mcp | Google Chrome | `open_profile_settings`
Opens the profile settings page in the browser |
| 31 | osworld_mcp | Google Chrome | `open_search_engine_settings`
Opens the search engine settings page in the browser |
| 32 | osworld_mcp | Google Chrome | `print`
Opens the print dialog for the current browser page (equivalent to Ctrl+P) |
| 33 | osworld_mcp | LibreOffice Calc | `adjust_column_width`
Adjust the width of specified columns |
| 34 | osworld_mcp | LibreOffice Calc | `adjust_row_height`
Adjust the height of specified rows |
| 35 | osworld_mcp | LibreOffice Calc | `copy_sheet`
Create a copy of an existing worksheet in the workbook |
| 36 | osworld_mcp | LibreOffice Calc | `create_chart`
Create a chart in the active worksheet based on the specified data range |
| 37 | osworld_mcp | LibreOffice Calc | `create_pivot_table`
Create a pivot table in the active worksheet based on data from the source sheet |
| 38 | osworld_mcp | LibreOffice Calc | `env_info`
Get content of the specified or active sheet, including its name, headers, and data |
| 39 | osworld_mcp | LibreOffice Calc | `export_to_csv`
Export the current document to a CSV file with the same path and name as the original file |
| 40 | osworld_mcp | LibreOffice Calc | `export_to_pdf`
Export the current document or specified sheets to PDF |
| 41 | osworld_mcp | LibreOffice Calc | `format_range`
Apply formatting to the specified range in the active worksheet |
| 42 | osworld_mcp | LibreOffice Calc | `freeze_panes`
Freeze rows and/or columns in the active worksheet |
| 43 | osworld_mcp | LibreOffice Calc | `get_column_data`
Get all data from the specified column |
| 44 | osworld_mcp | LibreOffice Calc | `get_workbook_info`
Get workbook information, including file path, file name, sheets and active sheet |
| 45 | osworld_mcp | LibreOffice Calc | `hide_row_data`
Hide rows that contain the specified value |
| 46 | osworld_mcp | LibreOffice Calc | `highlight_range`
Highlight the specified range with the specified color |
| 47 | osworld_mcp | LibreOffice Calc | `merge_cells`
Merge cells in the specified range |
| 48 | osworld_mcp | LibreOffice Calc | `rename_sheet`
Rename a worksheet in the workbook |
| 49 | osworld_mcp | LibreOffice Calc | `reorder_columns`
Reorder the columns in the sheet according to the specified order |

| No. | MCP Server | Domain | MCP Tool |
|---|---|---|---|
| 50 | osworld_mcp | LibreOffice Calc | `reorder_sheets`
Change the order of worksheets in the workbook |
| 51 | osworld_mcp | LibreOffice Calc | `save`
Save the current workbook to its current location |
| 52 | osworld_mcp | LibreOffice Calc | `set_cell_value`
Set a value to a specific cell in the active worksheet |
| 53 | osworld_mcp | LibreOffice Calc | `set_column_values`
Set values to the specified column, cannot be used to set formulas |
| 54 | osworld_mcp | LibreOffice Calc | `set_number_format`
Apply a specific number format to a range of cells |
| 55 | osworld_mcp | LibreOffice Calc | `set_zoom_level`
Adjust the zoom level of the current worksheet |
| 56 | osworld_mcp | LibreOffice Calc | `sort_column`
Sort the data in the specified column in ascending or descending order |
| 57 | osworld_mcp | LibreOffice Calc | `switch_active_sheet`
Switch to the specified sheet and make it active. Creates new sheet if it doesn't exist |
| 58 | osworld_mcp | LibreOffice Calc | `transpose_range`
Transpose the specified range and paste it to the target cell |
| 59 | osworld_mcp | LibreOffice Calc | `copy_cells_between_sheets`
Copy cells from a specified rectangular source range to another sheet |
| 60 | osworld_mcp | LibreOffice Calc | `fill_blank_down`
Forward-fills blank cells in specified columns with value from cell above |
| 61 | osworld_mcp | LibreOffice Calc | `format_numbers_to_human_readable`
Convert numeric values to human-readable strings (M for millions, B for billions) |
| 62 | osworld_mcp | LibreOffice Calc | `scale_first_sheet_and_export_pdf`
Scales the first sheet to specified pages and exports to PDF |
| 63 | osworld_mcp | LibreOffice Writer | `save`
Save the current document to its current location |
| 64 | osworld_mcp | LibreOffice Writer | `write_text`
Write text at the current cursor position in the document |
| 65 | osworld_mcp | LibreOffice Writer | `set_color`
Changes the color of matched text in the document |
| 66 | osworld_mcp | LibreOffice Writer | `find_and_replace`
Finds and replaces text in the document |
| 67 | osworld_mcp | LibreOffice Writer | `set_font`
Changes the font of text in the document |
| 68 | osworld_mcp | LibreOffice Writer | `set_line_spacing`
Sets the line spacing for specified paragraphs |
| 69 | osworld_mcp | LibreOffice Writer | `insert_formula_at_cursor`
Inserts a formula at the current cursor position |
| 70 | osworld_mcp | LibreOffice Writer | `insert_image_at_cursor`
Inserts an image at the current cursor position |

| No. | MCP Server | Domain | MCP Tool |
|-----|-----------|--------|----------|
| 71 | osworld_mcp | LibreOffice Writer | `set_font_size` 
 Changes the font size of specified text |
| 72 | osworld_mcp | LibreOffice Writer | `export_to_pdf` 
 Exports the current document to PDF format |
| 73 | osworld_mcp | LibreOffice Writer | `set_paragraph_alignment` 
 Sets the text alignment for specified paragraphs |
| 74 | osworld_mcp | LibreOffice Writer | `set_default_font` 
 Sets the default font for new text without changing existing text |
| 75 | osworld_mcp | LibreOffice Writer | `add_page_numbers` 
 Adds page numbers to the document at specified position |
| 76 | osworld_mcp | LibreOffice Writer | `insert_page_break` 
 Inserts a page break at current cursor position |
| 77 | osworld_mcp | LibreOffice Writer | `env_info` 
 Retrieve all paragraphs, truncate each to at most 500 characters |
| 78 | osworld_mcp | LibreOffice Impress | `configure_auto_save` 
 Enables or disables auto-save functionality |
| 79 | osworld_mcp | LibreOffice Impress | `delete_content` 
 Deletes the specified textbox from a slide |
| 80 | osworld_mcp | LibreOffice Impress | `duplicate_slide` 
 Creates a duplicate of a specific slide |
| 81 | osworld_mcp | LibreOffice Impress | `env_info` 
 Get the content of the specified pages |
| 82 | osworld_mcp | LibreOffice Impress | `export_to_image` 
 Exports the current presentation or a specific slide to an image file |
| 83 | osworld_mcp | LibreOffice Impress | `get_slide_count` 
 Gets the total number of slides in the current presentation |
| 84 | osworld_mcp | LibreOffice Impress | `go_to_slide` 
 Navigates to a specific slide in the presentation |
| 85 | osworld_mcp | LibreOffice Impress | `insert_file` 
 Insects a video or audio file into the current or specified slide |
| 86 | osworld_mcp | LibreOffice Impress | `insert_image` 
 Inserts an image to a specific slide |
| 87 | osworld_mcp | LibreOffice Impress | `position_box` 
 Positions a textbox or image on a slide at a specific location |
| 88 | osworld_mcp | LibreOffice Impress | `save` 
 Save the current presentation to its current location |
| 89 | osworld_mcp | LibreOffice Impress | `save_as` 
 Saves the current document to a specified location |
| 90 | osworld_mcp | LibreOffice Impress | `set_background_color` 
 Sets the background color for the specified textbox |
| 91 | osworld_mcp | LibreOffice Impress | `set_slide_background` 
 Sets the background color or image for a specific slide or all slides |
| 92 | osworld_mcp | LibreOffice Impress | `set_slide_font` 
 Sets the font style for all text elements in a specific slide |

| No. | MCP Server | Domain | MCP Tool |
|---|---|---|---|
| 93 | osworld_mcp | LibreOffice Impress | `set_slide_orientation`
Changes the orientation of slides between portrait and landscape |
| 94 | osworld_mcp | LibreOffice Impress | `set_style`
Sets the style properties for the specified textbox |
| 95 | osworld_mcp | LibreOffice Impress | `set_text_color`
Sets the text color for the specified textbox |
| 96 | osworld_mcp | LibreOffice Impress | `set_text_strikethrough`
Applies or removes strike-through formatting to text |
| 97 | osworld_mcp | LibreOffice Impress | `set_textbox_alignment`
Sets the text alignment for the specified textbox |
| 98 | osworld_mcp | LibreOffice Impress | `write_text`
writes text to a specific textbox on a slide |
| 99 | osworld_mcp | LibreOffice Impress | `convert_to_docx`
Transfers all text slide-by-slide into a new Writer document |
| 100 | osworld_mcp | OS | `change_text_scale`
Changes the text-scaling factor and returns the previous value |
| 101 | osworld_mcp | OS | `configure_auto_lock`
Configures the GNOME automatic-lock behaviour |
| 102 | osworld_mcp | OS | `copy_matching_files_with_hierarchy`
Copies all files matching a pattern preserving directory hierarchy |
| 103 | osworld_mcp | OS | `get_do_not_disturb_status`
Returns the current Do Not Disturb state |
| 104 | osworld_mcp | OS | `get_text_scale`
Returns the current GNOME text-scaling factor |
| 105 | osworld_mcp | OS | `get_trash_directory`
Returns the absolute path to the user Trash 'files' directory |
| 106 | osworld_mcp | OS | `get_volume`
Returns the current output volume (percentage) |
| 107 | osworld_mcp | OS | `open_shell`
Open a new terminal directly |
| 108 | osworld_mcp | OS | `rename_directory`
Safely renames a folder on the local filesystem |
| 109 | osworld_mcp | OS | `restore_file`
Restore the specified file from the user Trash back to its original location |
| 110 | osworld_mcp | OS | `search_files`
Recursively search all files under the given folder |
| 111 | osworld_mcp | OS | `set_default_terminal_size`
Persists the given number of columns × rows as the default GNOME-Terminal window size |
| 112 | osworld_mcp | OS | `set_do_not_disturb`
Enable or disable the GNOME/Ubuntu Do Not Disturb mode |

| No. | MCP Server | Domain | MCP Tool |
|---|---|---|---|
| 113 | osworld_mcp | OS | `set_volume`
Sets the output volume of the default PulseAudio / PipeWire sink |
| 114 | osworld_mcp | OS | `convert_image_format`
Convert an input image file to a specified format |
| 115 | osworld_mcp | OS | `ffmpeg_video_to_gif`
Extract a portion of a video file and save it as an animated GIF |
| 116 | osworld_mcp | OS | `git_operation`
Execute a git command (clone, add, commit, pull, push, etc.) |
| 117 | osworld_mcp | OS | `git_set_user_info`
Configure git user.name and user.email |
| 118 | osworld_mcp | OS | `remove_image_background`
Remove the background of an input image and save with transparency |
| 119 | osworld_mcp | OS | `calculator`
Simple calculator. |
| 120 | osworld_mcp | VLC | `add_to_playlist`
Adds a media file to the VLC playlist |
| 121 | osworld_mcp | VLC | `get_current_time`
Gets the current playback time position in seconds |
| 122 | osworld_mcp | VLC | `get_media_duration`
Gets the total duration of the currently playing media file in seconds |
| 123 | osworld_mcp | VLC | `get_media_files`
Gets the media files for the specified path |
| 124 | osworld_mcp | VLC | `get_playlist`
Gets the current VLC playlist with track information |
| 125 | osworld_mcp | VLC | `get_settings`
Gets the current settings of the VLC player |
| 126 | osworld_mcp | VLC | `next`
Switches to the next media item in the VLC playlist |
| 127 | osworld_mcp | VLC | `pause`
Pauses the currently playing media in VLC player |
| 128 | osworld_mcp | VLC | `play`
Starts playing the current media in VLC player |
| 129 | osworld_mcp | VLC | `previous`
Switches to the previous media item in the VLC playlist |
| 130 | osworld_mcp | VLC | `set_settings`
Sets the settings for the VLC player |
| 131 | osworld_mcp | VLC | `toggle_fullscreen`
Toggles fullscreen mode for the currently playing video |
| 132 | filesystem | OS | `read_file`
Read the complete contents of a file as text (DEPRECATED) |
| 133 | filesystem | OS | `read_text_file`
Read the complete contents of a file as text with encoding handling |

| No. | MCP Server | Domain | MCP Tool |
|-----|-----------|--------|----------|
| 134 | filesystem | OS | `read_media_file`
Read an image or audio file as base64 encoded data |
| 135 | filesystem | OS | `read_multiple_files`
Read the contents of multiple files simultaneously |
| 136 | filesystem | OS | `write_file`
Create a new file or completely overwrite an existing file |
| 137 | filesystem | OS | `edit_file`
Make line-based edits to a text file |
| 138 | filesystem | OS | `create_directory`
Create a new directory or ensure a directory exists |
| 139 | filesystem | OS | `list_directory`
Get a detailed listing of all files and directories in a specified path |
| 140 | filesystem | OS | `list_directory_with_sizes`
Get a detailed listing with sizes of all files and directories |
| 141 | filesystem | OS | `directory_tree`
Get a recursive tree view of files and directories as JSON |
| 142 | filesystem | OS | `move_file`
Move or rename files and directories |
| 143 | filesystem | OS | `search_files`
Recursively search for files and directories matching a pattern |
| 144 | filesystem | OS | `get_file_info`
Retrieve detailed metadata about a file or directory |
| 145 | filesystem | OS | `list_allowed_directories`
Returns the list of directories that this server is allowed to access |
| 146 | git | OS | `git_status`
Shows the working tree status |
| 147 | git | OS | `git_diff_unstaged`
Shows changes in the working directory that are not yet staged |
| 148 | git | OS | `git_diff_staged`
Shows changes that are staged for commit |
| 149 | git | OS | `git_diff`
Shows differences between branches or commits |
| 150 | git | OS | `git_commit`
Records changes to the repository |
| 151 | git | OS | `git_add`
Adds file contents to the staging area |
| 152 | git | OS | `git_reset`
Unstages all staged changes |
| 153 | git | OS | `git_log`
Shows the commit logs |
| 154 | git | OS | `git_create_branch`
Creates a new branch from an optional base branch |
| 155 | git | OS | `git_checkout`
Switches branches |

| No. | MCP Server | Domain | MCP Tool |
|-----|-----------|--------|----------|
| 156 | git | OS | `git_show` |
|     |           |    | Shows the contents of a commit |
| 157 | git | OS | `git_init` |
|     |           |    | Initialize a new Git repository |
| 158 | git | OS | `git_branch` |
|     |           |    | List Git branches |

## B  PROMPT

The following shows the complete prompt send to the LLM during task execution, together with the corresponding screenshot. The user's instruction is: Please help me modify VS Code setting to hide all "__pycache__" folders in the explorer view.

**System prompt:**
You are a helpful assistant.
You may call one or more functions to assist with the user query.

You are provided with function signatures within <tools></tools>XML tags:
<tools>

{"type": "function", "function": {"name": "computer_use", "description": "Use a mouse and keyboard to interact with a computer, and take screenshots.\n* This is an interface to a desktop GUI. You do not have access to a terminal or applications menu. You must click on desktop icons to start applications.\n* Some applications may take time to start or process actions, so you may need to wait and take successive screenshots to see the results of your actions. E.g. if you click on Firefox and a window doesn't open, try wait and taking another screenshot.\n* The screen's resolution is 1280x720.\n* Whenever you intend to move the cursor to click on an element like an icon, you should consult a screenshot to determine the coordinates of the element before moving the cursor.\n* If you tried clicking on a program or link but it failed to load, even after waiting, try adjusting your cursor position so that the tip of the cursor visually falls on the element that you want to click.\n* Make sure to click any buttons, links, icons, etc with the cursor tip in the center of the element. Don't click boxes on their edges unless asked.", "parameters": {"properties": {"action": {"description": "The action to perform. The available actions are:\n* 'key': Performs key down presses on the arguments passed in order, then performs key releases in reverse order.\n* 'type': Input a string of text. Use the 'clear' parameter to decide whether to overwrite the existing text, and use the 'enter' parameter to decide whether the enter key should be pressed after typing the text.\n* 'mouse_move': Move the cursor to a specified (x, y) pixel coordinate on the screen.\n* 'click': Click the left mouse button at a specified (x, y) pixel coordinate on the screen.\n* 'drag': Click at a specified (x, y) pixel coordinate on the screen, and drag the cursor to another specified (x2, y2) pixel coordinate on the screen.\n* 'right_click': Click the right mouse button at a specified (x, y) pixel coordinate on the screen.\n* 'middle_click': Click the middle mouse button at a specified (x, y) pixel coordinate on the screen.\n* 'double_click': Double-click the left mouse button at a specified (x, y) pixel coordinate on the screen.\n* 'scroll': Performs a scroll of the mouse scroll wheel.\n* 'wait': Wait specified seconds for the change to happen.\n* 'terminate': Terminate the current task and report its completion status.", "enum": ["key", "type", "mouse_move", "click", "drag", "right_click", "middle_click", "double_click", "scroll", "wait", "terminate"], "type": "string"}, "keys": {"description": "Required only by 'action=key'.", "type": "array"}, "text": {"description": "Required only by 'action=type'.", "type": "string"}, "clear": {"description": "Assign it to 1 if the text should overwrite the existing text, otherwise assign it to 0. Using this argument clears all text in an element. Required only by 'action=type'.", "type": "number"}, "enter": {"description": "Assign it to 1 if the enter key should be pressed after typing the text, otherwise assign it to 0. Required only by 'action=type'.", "type": "number"}, "coordinate": {"description": "(x, y): The x (pixels from the left edge) and y (pixels from the top edge) coordinates to move the mouse to.", "type": "array"}, "coordinate2": {"description": "(x2, y2):

The x2 (pixels from the left edge) and y2 (pixels from the top edge) coordinates to drag the cursor to. Required only by 'action=drag'.", "type": "array"}, "pixels": {"description": "The amount of scrolling to perform. Positive values scroll up, negative values scroll down. This value should be between -5 and 5. Required only by 'action=scroll'.", "type": "number"}, "time": {"description": "The seconds to wait. Required only by 'action=wait'.", "type": "number"}, "status": {"description": "The status of the task. Required only by 'action=terminate'.", "type": "string", "enum": ["success", "failure"]}}, "required": ["action"], "type": "object"}}}
{"type": "function", "function": {"name": "osworld_mcp_code.add_folder", "description": "Adds a folder to the last active window in VSCode", "parameters": {"type": "object", "properties": {"folder": {"type": "string", "description": "The folder path to add"}}, "required": ["folder"]}}}
{"type": "function", "function": {"name": "osworld_mcp_code.compare_files", "description": "Compares two files in VSCode", "parameters": {"type": "object", "properties": {"file1": {"type": "string", "description": "The path to the first file"}, "file2": {"type": "string", "description": "The path to the second file"}}, "required": ["file1", "file2"]}}}
{"type": "function", "function": {"name": "osworld_mcp_code.disable_extension", "description": "Disables a specific extension for the next instance of VSCode", "parameters": {"type": "object", "properties": {"extension_id": {"type": "string", "description": "The identifier of the extension"}}, "required": ["extension_id"]}}}
{"type": "function", "function": {"name": "osworld_mcp_code.goto_file", "description": "Opens a file at a specific line and character position", "parameters": {"type": "object", "properties": {"file_path": {"type": "string", "description": "The file path to open"}, "line": {"type": "integer", "description": "The line number to navigate to", "default": 1}, "character": {"type": "integer", "description": "The character position to navigate to", "default": 1}}, "required": ["file_path"]}}}
{"type": "function", "function": {"name": "osworld_mcp_code.install_extension", "description": "Installs an extension or updates it in VSCode", "parameters": {"type": "object", "properties": {"extension_id": {"type": "string", "description": "The identifier of the extension"}, "pre_release": {"type": "boolean", "description": "Whether to install the pre-release version", "default": false}}, "required": ["extension_id"]}}}
{"type": "function", "function": {"name": "osworld_mcp_code.launch_vscode", "description": "Launches Visual Studio Code with the specified file path or directory", "parameters": {"type": "object", "properties": {"path": {"type": "string", "description": "The file path or directory to open in VS Code"}}, "required": ["path"]}}}
{"type": "function", "function": {"name": "osworld_mcp_code.list_extensions", "description": "Lists installed extensions in VSCode", "parameters": {"type": "object", "properties": {"show_versions": {"type": "boolean", "description": "Whether to show extension versions", "default": false}, "category": {"type": "string", "description": "The category to filter extensions by"}}}}}
{"type": "function", "function": {"name": "osworld_mcp_code.remove_folder", "description": "Removes a folder from the last active window in VSCode", "parameters": {"type": "object", "properties": {"folder": {"type": "string", "description": "The folder path to remove"}}, "required": ["folder"]}}}
{"type": "function", "function": {"name": "osworld_mcp_code.toggle_sync", "description": "Toggles synchronization on or off in VSCode", "parameters": {"type": "object", "properties": {"state": {"type": "string", "description": "The state to set ('on' or 'off')", "enum": ["on", "off"]}}, "required": ["state"]}}}
{"type": "function", "function": {"name": "osworld_mcp_code.uninstall_extension", "description": "Uninstalls an extension from VSCode", "parameters": {"type": "object", "properties": {"extension_id": {"type": "string", "description": "The identifier of the extension"}}, "required": ["extension_id"]}}}
{"type": "function", "function": {"name": "osworld_mcp_code.update_extensions", "description": "Updates all installed extensions in VSCode to the latest version", "parameters": {"type": "object", "properties": {}}}}
{"type": "function", "function": {"name": "osworld_mcp_code.add_files_exclude", "description": "Add a glob pattern to the Files: Exclude setting.", "parameters": {"type": "object", "properties": {"pattern": {"type": "string", "description": "Glob pattern to exclude, e.g.

'**/__pycache__'."}}, "required": ["pattern"]}}}
{"type": "function", "function": {"name": "osworld_mcp_code.replace_text", "description": "Open VSCode search and replace panel, input search and replacement text, and execute replacement in either all files or current file.", "parameters": {"type": "object", "properties": {"search_text": {"type": "string", "description": "The text to search for."}, "replace_text": {"type": "string", "description": "The text to replace matches with."}, "all_files": {"type": "boolean", "description": "True to replace in all files, false to replace only in the current file. Default is True."}}, "required": ["search_text", "replace_text"]}}}
{"type": "function", "function": {"name": "osworld_mcp_code.search_text", "description": "Open VSCode search panel, input the search term, and execute search in either all files or current file.", "parameters": {"type": "object", "properties": {"text": {"type": "string", "description": "The text to search for."}, "all_files": {"type": "boolean", "description": "True to search in all files, false to search only in the current file. Default is True."}}, "required": ["text"]}}}
{"type": "function", "function": {"name": "osworld_mcp_code.set_auto_save_delay", "description": "Set the auto save delay in milliseconds.", "parameters": {"type": "object", "properties": {"delay_ms": {"type": "integer", "description": "Delay before auto save in milliseconds."}}, "required": ["delay_ms"]}}}
{"type": "function", "function": {"name": "osworld_mcp_code.set_color_theme", "description": "Change the editor color theme.", "parameters": {"type": "object", "properties": {"theme_name": {"type": "string", "description": "The name of the theme to apply, e.g. 'Default Dark+'."}}, "required": ["theme_name"]}}}
{"type": "function", "function": {"name": "osworld_mcp_code.set_focus_editor_on_break", "description": "Set the Debug: Focus Editor On Break setting to true or false.", "parameters": {"type": "object", "properties": {"value": {"type": "boolean", "description": "True to enable focus on editor when debugger breaks, false to disable."}}, "required": ["value"]}}}
{"type": "function", "function": {"name": "osworld_mcp_code.set_python_diagnostics_override", "description": "Override severity for a specific Python analysis diagnostic rule.", "parameters": {"type": "object", "properties": {"rule": {"type": "string", "description": "Diagnostic rule ID, e.g. 'reportMissingImports'."}, "severity": {"type": "string", "description": "Severity level, e.g. 'info', 'warning', 'error', or 'none'."}}, "required": ["rule", "severity"]}}}
{"type": "function", "function": {"name": "osworld_mcp_code.set_word_wrap_column", "description": "Set the number of columns at which the editor will word wrap.", "parameters": {"type": "object", "properties": {"column": {"type": "integer", "description": "The column number where lines should wrap, e.g. 80 or 120."}}, "required": ["column"]}}}
{"type": "function", "function": {"name": "osworld_mcp_code.set_wrap_tabs", "description": "Enable or disable wrap tabs in the editor.", "parameters": {"type": "object", "properties": {"value": {"type": "boolean", "description": "True to enable wrap tabs, false to disable."}}, "required": ["value"]}}}

</tools>

For each function call, return a json object with function name and arguments within <tool_call></tool_call>XML tags:
<tool_call>
{"name": <function-name>, "arguments": <args-json-object>}

</tool_call>

**User Prompt**
Please generate the next move according to the UI screenshot, instruction and previous actions.\n Instruction: Please help me modify VS Code setting to hide all "__pycache__" folders in the explorer view.
**Previous actions:**
- **Step 1**: I first clicked "Yes, I trust the authors" to dismiss the trust dialog, then used the 'add_files_exclude' function to add the glob pattern "**/__pycache__" to VS Code's

Files: Exclude setting. This pattern will hide all __pycache__ folders at any depth in the project structure from the explorer view.

**Tool call result:**

```
CallToolResult(content=[TextContent(type='text', text='true',
↪  annotations=None, meta=None)], structured_content=None,
↪  data=None, is_error=False)
```

Before answering, explain your reasoning step-by-step in <thinking></thinking>tags, and insert them before the <tool_call></tool_call>XML tags.\nAfter answering, summarize your action in <conclusion></conclusion>tags, and insert them after the <tool_call></tool_call>XML tags.

**History Images**

SCREENSHOT OF STEP 1:

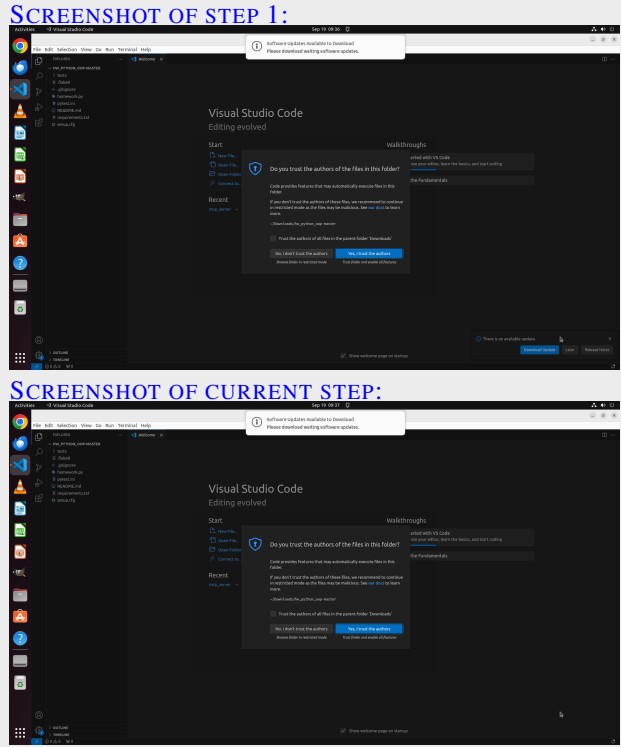

SCREENSHOT OF CURRENT STEP:

## B.1 USE OF LLMS

Large language models (LLMs) are used solely to assist in the preparation of this manuscript. They help improve the clarity, coherence, and conciseness of the text, refine phrasing, and ensure that the language conforms to academic writing standards. All conceptual content, experimental design, data analysis, and conclusions are developed entirely by the authors.

