# OpenReview forum: "OSWorld-MCP: Benchmarking MCP Tool Invocation In Computer-Use Agents"
_ICLR.cc/2026/Conference — ICLR 2026 Poster_

### Official Review · Reviewer_DEYi · 2025-10-28

**Soundness:** 3
**Presentation:** 3
**Contribution:** 3
**Rating:** 6
**Confidence:** 4

**Summary:**

This paper presents OSWorld-MCP, a novel and comprehensive benchmark framework to evaluate computer-use agents within the OSWorld benchmark. This framework incorporates tool use (via the Model Context Protocol, or MCP) into the standard GUI benchmark. The authors show that by leveraging the MCP tools, agents achieve improved task success in OSWorld tasks.
This paper also introduces an end-to-end pipeline to curate MCP tools, leveraging both LLM tool generation and human verification. The paper presents 158 high-quality, manually-validated MCP tools, allowing agents to autonomously choose between GUI operations and tool calls.

**Strengths:**

This is a solid, well-written paper. The experimental design is very detailed, and the authors do a great job discussing the results, which are supported by clear visualizations. I especially appreciated the ablation studies on the number of tools, the RAG module, and the order of tools in the prompt—these were very interesting. The detailed case studies in the appendix are also a helpful addition.

The core idea of combining MCP tools with a GUI agent is novel. This is a strong piece of work that should be a valuable contribution to research on both MCP and GUI agents.

**Weaknesses:**

I would like to see more detail on the RAG module's design for the MCP tool selection. The paper doesn't specify what retrieval method was used or its retrieval accuracy, which makes it hard to fully understand that component of the experiment. Additionally, the tools were generated by "producing code-based solutions for every task in OSWorld", which may cause an unfair comparison. Since the tools were created using the same tasks they were evaluated on, there's a risk of task information leakage. It would be better to generate the tools using one set of tasks and then evaluate their performance on a separate, held-out validation set.

**Questions:**

See above

---

> ### Author Response · Authors · 2025-11-28
> **Rebuttal by Authors: Q1, Q2**
>
> We sincerely thank you for your thoughtful feedback and for your *weakly accept* recommendation. Your comments have been valuable in helping us clarify and improve our work. Below, we provide detailed responses to the specific points you raised.
>
> ---
>
> ## Q1: Details on RAG
>
> **Reviewer'Comment:**  I would like to see more detail on the RAG module's design for the MCP tool selection. The paper doesn't specify what retrieval method was used or its retrieval accuracy, which makes it hard to fully understand that component of the experiment.
>
> **Our Response:**
>
> Thank you for your insightful question. Using RAG to select the appropriate tool plays an important role in MCP tool invocation.
>
> 1. **Retrieval Pipeline Design**
> Our RAG module employs a lightweight yet effective pipeline.
> **Step 1:** Detect the software currently in use by monitoring the active application context within the OSWorld environment.
> **Step 2:** Filter MCP tools corresponding to that software from our curated tool set. If no software is currently open, OS‑related MCP tools are provided instead.
> **Step 3:** Within the reduced candidate set, retrieve relevant tools using simple keyword‑based matching on tool names and descriptions. This approach avoids overfitting to training data and maintains interpretability in retrieval.
>
> 2. **Retrieval Performance and Analysis**
> As shown in Table 2 of the paper, after applying our RAG module, the accuracy of Gemini 2.5 Pro **increased by 32.3%**. Our preliminary experiments indicate that this simple filtering method, combined with keyword‑based retrieval, is sufficient to reliably return the correct tools without imposing substantial computational overhead as observed with vector‑based search methods. Furthermore, it keeps the selection process transparent, aligning with our objectives of fairness and reproducibility in tool invocation evaluation. **Using RAG to select appropriate tools represents an important research direction for enhancing the capabilities of computer‑use agents in the future.**
>
> ---
>
> ## Q2: The impact of tools on fairness
>
> **Reviewer'Comment:**  Additionally, the tools were generated by "producing code-based solutions for every task in OSWorld", which may cause an unfair comparison. Since the tools were created using the same tasks they were evaluated on, there's a risk of task information leakage. It would be better to generate the tools using one set of tasks and then evaluate their performance on a separate, held-out validation set.
>
> **Our Response:**
>
> Regarding the task leakage risk you raised, we have taken this risk into account during the manual verification stage.
>
> Given that external MCP tools and tools generated through the automated tool generation pipeline may pose potential task leakage risks, as stated in the original paper, we performed **detailed manual verification** on each collected tool. We removed tools with overlapping functionalities or **those that could lead to task leakage**. For example, the perform_merge tool, which carries out a “three‑way merge” function, was excluded because its functionality almost directly overlaps with certain OSWorld tasks and has limited practical utility in real‑world scenarios. All 158 tools in OSWorld‑MCP were manually verified to **ensure task fairness**.
>
> ---
>
> We truly appreciate your constructive comments. Your feedback has helped us further refine both our methodology and explanation, and we believe the clarifications above strengthen the paper’s contributions. Thank you again for your careful reading and valuable suggestions.

---

### Official Review · Reviewer_SQsy · 2025-11-01

**Soundness:** 2
**Presentation:** 2
**Contribution:** 2
**Rating:** 2
**Confidence:** 3

**Summary:**

The paper introduces OSWorld-MCP, a benchmark designed to fairly and comprehensively evaluate multimodal computer-use agents that combine GUI interaction with Model Context Protocol (MCP)–based tool invocation. It extends OSWorld by integrating 158 manually verified MCP tools covering 7 common applications (LibreOffice, VS Code, Chrome, VLC, OS utilities, etc.).

**Strengths:**

- The automated pipeline (generation, filtering, wrapping) for tool creation is well-structured and uses both LLM-based and manual verification steps.

- Covers multiple proprietary and open models under consistent settings, providing valuable comparative insights.

**Weaknesses:**

- While benchmark creation idea is strong, analysis mostly confirms intuitive findings (e.g., higher TIR means higher accuracy)

- The paper’s main contribution (augmenting OSWorld with MCP tool invocation) is meaningful, but I’m concerned that OSWorld’s task design predates MCP and may not naturally align with tool-based workflows. The authors partly mitigate this by revalidating 361 tasks, identifying 250 “tool-beneficial” cases, and manually curating 158 tools, which shows care in adaptation. However, the benchmark still fundamentally inherits OSWorld’s task definitions, rather than introducing new ones designed from the ground up for MCP-enabled agents. This limits its novelty and may constrain how representative it is of modern tool-use scenarios

- Some results (Tables 1–2) lack significance testing or variance measures; performance improvements could benefit from confidence intervals.

- While clear overall, the draft is verbose in places and could improve figure readability and section organization. E.g. Figure 1 contained red wavy dash, need fix.

**Questions:**

1. How was task difficulty balanced between GUI-only and MCP-enhanced versions to ensure fairness?

2. Could tool invocation failures be categorized (e.g., recognition vs. reasoning vs. execution errors)?

---

> ### Author Response · Authors · 2025-11-28
> **Rebuttal by Authors: W1-Insight1**
>
> We thank the reviewer for the time and effort spent reading our submission and for the feedback provided. We address the main concerns below and clarify several potential misunderstandings regarding our contributions, methodology, and evaluation results.
>
> ## W1: Intuitive Findings
>
> **Reviewer'Comment:**  While benchmark creation idea is strong, analysis mostly confirms intuitive findings (e.g., higher TIR means higher accuracy)
>
> **Our Response:**
>
> We appreciate the reviewer’s comment but respectfully disagree that our results merely confirm intuitive relationships. To address this, we have **reformulated the four key insights from the original paper and provided additional in‑depth analyses during the rebuttal**, as detailed in the following sections. The novelty and rigor of our methodology, benchmark design, and analyses have also been explicitly recognized by other reviewers (8XpA, cXcb, DEYi), reinforcing that our contributions extend well beyond confirming intuitive trends.
>
> **1. Correct tool usage is rare but critical**
>
> As shown in **Table 1**, even after introducing MCP tools, the **Tool Invocation Rate (TIR)** for the most advanced models remains only **36.3%**, which **substantially limits the potential improvement** in model performance.
>
> To further verify the impact of MCP tools on overall capability, we randomly sampled **200 tasks** from those involving tool invocation for detailed analysis. These tasks were divided into two categories:
> - Correct tool use tasks in which the model invoked the **correct tool**
> - Incorrect tool use tasks in which the model invoked the **incorrect tool**
>
> The statistical results are shown below:
>
> | Task Category             | Correct tool use tasks | Incorrect tool use tasks | Total |
> |---------------------------|-------------------------|---------------------------|-------|
> | Number of tasks           | 108                      | 92                        | 200    |
> | Accuracy (%)              | 51.9                    | 8.7                       | 32.0 |
>
> **Analysis:**
>
> The proportion of tasks with correct tool invocation is **only 54% (108/200)**. When tools are **invoked correctly**, the model’s ability to solve computer‑use tasks improves **dramatically**. Conversely, when tools are invoked **incorrectly** during execution, accuracy drops **sharply**. When tools are **invoked correctly**, task success rate improves by  nearly more than **6×** compared to incorrect invocation (51.9% vs 8.7%).
> This shows that **the main performance bottleneck is the ability to select and use the right tool**. Improving this capability could yield disproportionately large gains, and the OSWorld‑MCP benchmark surfaces this bottleneck clearly.

---

> ### Author Response · Authors · 2025-11-28
> **Rebuttal by Authors: W1-Insight2 & Insight3 & Statistical Confirmation**
>
> **2. Failure tool usage mainly comes from three errors: Tool‑misuse errors, GUI‑tool integration errors, and GUI‑overreliance errors.**
> For **250 Tool‑Beneficial tasks that require invoking tools**, we performed a detailed error analysis across Qwen2.5-VL, OpenAI o3, Claude 4 Sonnet, and identifies three dominant error types:
> (1) **Tool‑misuse errors:** selecting the wrong tool type or parameters, revealing weaknesses in reasoning and task‑to‑tool mapping.
> (2) **GUI‑tool integration errors:** invoking the correct MCP tool but failing to integrate its results with GUI operations, exposing shortcomings in task decomposition.
> (3) **GUI‑overreliance errors:** relying solely on GUI actions and failing to recognize tool‑required tasks.
>
> The error distribution is as follows:
>
> | Model           | GUI‑overreliance errors | Tool‑misuse errors | GUI‑tool integration errors | Total |
> |-----------------|-------------------------|--------------------|------------------------------|-------|
> | Qwen2.5‑VL      | 69                      | 67                 | 81                           | 217   |
> | OpenAI o3       | 32                      | 63                 | 67                           | 162   |
> | Claude 4 Sonnet | 21                      | 57                 | 33                           | 111   |
>
> **Key findings:**
> (1) **Tool‑misuse** errors occur most frequently, indicating that **improving tool‑selection logic should be a priority** for advancing performance.
> (2) **GUI‑overreliance** and **GUI‑tool integration** errors tend to decrease in more capable models, but weaker ones (e.g., Qwen2.5‑VL) still struggle to make accurate “routing” decisions—whether to invoke a tool or proceed via GUI. This suggests that **explicit routing mechanisms during inference** could be effective.
> (3) **GUI‑tool integration errors** are more common in tasks with **multiple subtasks**. For example, in “Give the slide a purple background color. Add the title to note,” even advanced models (e.g., Claude 4 Sonnet) sometimes omit steps (“Add the title to note”) or mis‑order actions. This indicates that **global task comprehension** remains a challenge. We recommend two improvement directions:
> - Enrich agents’ world knowledge to better understand complex, real‑world tasks.
> - Improve **global planning capability** to mitigate limitations from restricted global context.
>
> ---
>
> **3. Multi‑round tool invocation poses a greater challenge than single‑round invocation and exposes severe global‑context limitations**
>
> Figure 5.(b) reveals the challenges posed to models by tasks requiring **multiple round tool invocation**. Specifically, compared with tasks that require only a single round of tool invocation, accuracy drops significantly **from 44.7% to 16.7%** when multiple rounds are needed, indicating poor multi‑round tool invocation capability.
> Our error analysis links this to weak **global context awareness**, which means agents often lose intermediate outputs (e.g., slide IDs, detected attributes) and fail to incorporate them into subsequent steps.
> This indicates a need for dedicated **context‑retention mechanisms** and **global‑planning modules** that persist task states across steps. It challenges that **would remain invisible without a benchmark incorporating multi‑round tool use**.
>
> ---
>
> - **Statistical Confirmation**: Another important purpose of building the benchmark is to use systematic experiments to verify or quantify cognitive assumptions that would otherwise rely on heuristics. As shown in Figure 5(a), although the positive correlation between TIR and task accuracy might seem intuitive, our conclusions are **derived from precise statistical computation** rather than heuristic speculation. Specifically, we report a Pearson correlation coefficient of r = 0.952 (p‑value = 3.62 × 10⁻²²) and a Spearman rank correlation coefficient of ρ = 0.955 (p‑value = 8.84 × 10⁻²³), both indicating a statistically significant positive correlation.
>
> We will include these additional analyses and improvement suggestions in the revised paper to address the reviewer’s concerns.

---

> ### Author Response · Authors · 2025-11-28
> **Rebuttal by Authors: W2**
>
> ## W2: Task Design Alignment
>
> **Reviewer'Comment:**  The paper’s main contribution (augmenting OSWorld with MCP tool invocation) is meaningful, but I’m concerned that OSWorld’s task design predates MCP and may not naturally align with tool-based workflows. The authors partly mitigate this by revalidating 361 tasks, identifying 250 “Tool-Beneficial” cases, and manually curating 158 tools, which shows care in adaptation. However, the benchmark still fundamentally inherits OSWorld’s task definitions, rather than introducing new ones designed from the ground up for MCP-enabled agents. This limits its novelty and may constrain how representative it is of modern tool-use scenarios.
>
> **Our Response:**
>
> 1. **Appropriateness of OSWorld as the Task Foundation**
> OSWorld tasks cover diverse, realistic computer-use scenarios with clear evaluation procedures, and are the most widely adopted GUI benchmark (Claude Opus 4.5, Qwen3‑VL, and OpenAI CUA are all evaluated using it).
> Their realism and broad applicability ensure comparability with prior work and relevance to actual agent usage.
>
> 2. **Ensuring MCP Workflow Alignment**
> As described in Section 3.2, we revalidated all 361 OSWorld tasks and **identify 250 “Tool-Beneficial” tasks** through systematic MCP suitability analysis.
> Using a pipeline that combines automated code generation and expert curation, we integrated 158 high-quality MCP tools directly mapped to these tasks, **ensuring strong tool–task alignment**.
> By integrating these MCP tools into all OSWorld task, we **redefine the evaluation paradigm** for GUI-based benchmarks, which advances evaluation beyond pure GUI manipulation to encompass **integrated reasoning across GUI operations and MCP tool usage**.
>
> 3. **Novel Contributions on Top of OSWorld**
> Our key innovations include:
>   - **MCP tool invocation evaluation:** First benchmark to assess agents’ ability to use MCP tools within GUI tasks, a skill increasingly critical for AI-native applications (Hou et al., 2025; Hasan et al., 2025; Chhetri et al., 2025).
>   - **Fair, integrated benchmarking:** Jointly evaluates tool use, GUI operation skills, and decision-making, addressing inequities in comparing GUI-only and API-equipped agents (Lai et al., 2025; Gonzalez-Pumariega et al., 2025; Song et al., 2025).
>   -**Automated tool generation pipeline:** Combines automated code generation and manual curation to create high-quality MCP tools, broadly applicable for future benchmarks.
>   - **New metrics:** Tool Invocation Rate (TIR) for measuring tool use propensity and Average Completion Steps (ACS) for task efficiency.

---

> ### Author Response · Authors · 2025-11-28
> **Rebuttal by Authors: W3**
>
> ## W3: Performance Improvements
>
> **Reviewer'Comment:**  Some results (Tables 1–2) lack significance testing or variance measures; performance improvements could benefit from confidence intervals.
>
> **Our Response:**
>
> Thank you for your suggestion. We ran each experiment **3 times** to ensure both reproducibility and statistical robustness of the results.
>
> - **Significance testing:** To verify whether the results of the same model on OSWorld‑MCP can represent the actual performance of the agents, we conducted a Wilcoxon signed‑rank test. For various agents, we performed two independent experiments, and based on 361 tasks, we carried out the Wilcoxon signed‑rank test. The experimental results are as follows:
>
> | Model           | Test statistic | p‑value | Difference significant (p < 0.05) |
> |-----------------|----------------|---------|-----------------------------------|
> | Qwen2.5‑VL      | 252.0          | 0.38    | No                                |
> | Qwen3‑VL        | 711.0          | 0.44    | No                                |
> | OpenAI o3       | 192.5          | 0.23    | No                                |
> | Gemini‑2.5‑Pro  | 168.0          | 0.56    | No                                |
> | Seed1.5‑VL      | 442.0          | 0.68    | No                                |
> | Claude‑4‑Sonnet | 865.0          | 0.52    | No                                |
> | Agent‑S2.5      | 727.0          | 0.12    | No                                |
>
>
> **Analysis:**
>
> The experimental results show that after introducing MCP, the differences between results from different runs under the same settings become smaller, and the models exhibit a high level of stability.
>
> - **Variance measures and confidence intervals:** To further **accurately evaluate** the performance of each agent, we conducted three independent runs for each agent. The table below reports **the mean, standard deviation, and 95% confidence interval** of **Accuracy, TIR, and ACS** for each model under different settings, with a maximum step count of 15. The complete table can be found in Appendix A.3.
>
>
> | Agent | Steps | Method | Accuracy (mean±std) | Accuracy (CI) | TIR (mean±std) | TIR (CI) | ACS (mean±std) | ACS (CI) |
> |----------|------|------|---------------------|---------------|----------------|----------|----------------|----------|
> | Agent S2.5 | 15 | GUI Only | 36.7 ± 2.7 | (33.6, 39.7) | - | - | 11.3 ± 0.1 | (11.1, 11.5) |
> | Agent S2.5 | 15 | GUI+MCP | 42.1 ± 1.5 | (40.4, 43.9) | 30.0 ± 1.4 | (28.4, 31.6) | 10.0 ± 0.2 | (9.8, 10.2) |
> | Seed-vl | 15 | GUI Only | 27.9 ± 1.3 | (26.4, 29.4) | - | - | 10.9 ± 0.0 | (10.8, 10.9) |
> | Seed-vl| 15 | GUI+MCP | 32.0 ± 0.3 | (31.6, 32.4) | 25.1 ± 0.1 | (25.0, 25.2) | 10.2 ± 0.1 | (10.1, 10.3) |
> | Qwen3-VL | 15 | GUI Only | 25.4 ± 1.3 | (23.8, 26.9) | - | - | 11.6 ± 0.1 | (11.5, 11.7) |
> | Qwen3-VL | 15 | GUI+MCP | 31.3 ± 1.8 | (29.2, 33.4) | 24.5 ± 2.2 | (22.0, 27.1) | 10.5 ± 0.2 | (10.3, 10.7) |
> | Claude 4 Sonnet | 15 | GUI Only | 30.2 ± 1.7 | (28.3, 32.1) | - | - | 11.9 ± 0.1 | (11.8, 12.0) |
> | Claude 4 Sonnet | 15 | GUI+MCP | 35.3 ± 2.3 | (32.7, 37.9) | 30.0 ± 1.7 | (28.1, 31.9) | 10.4 ± 0.1 | (10.2, 10.6) |
> | Gemini-2.5 Pro | 15 | GUI Only | 7.4 ± 3.1 | (3.9, 10.9) | - | - | 13.8 ± 0.5 | (13.3, 14.3) |
> | Gemini-2.5 Pro | 15 | GUI+MCP | 20.5 ± 2.5 | (17.8, 23.3) | 16.8 ± 0.2 | (16.6, 17.1) | 11.3 ± 0.4 | (10.9, 11.8) |
> | OpenAI o3 | 15 | GUI Only | 8.3 ± 3.8 | (4.1, 12.6) | - | - | 14.0 ± 0.2 | (13.8, 14.2) |
> | OpenAI o3 | 15 | GUI+MCP | 20.4 ± 1.4 | (18.8, 21.9) | 16.7 ± 0.7 | (15.9, 17.5) | 11.6 ± 0.4 | (11.2, 12.0) |
> | Qwen2.5-VL | 15 | GUI Only | 11.4 ± 1.8 | (9.4, 13.5) | - | - | 13.0 ± 0.1 | (12.8, 13.1) |
> | Qwen2.5-VL | 15 | GUI+MCP | 15.8 ± 0.5 | (15.3, 16.4) | 13.1 ± 2.2 | (10.6, 15.6) | 13.5 ± 0.2 | (13.3, 13.7) |
>
>
> **Analysis:**
>
>  The experimental results show that in most cases, after introducing MCP, the performance differences between runs under the same settings become smaller, and the models demonstrate greater stability. In certain cases, this may be related to the relatively low Tool Invocation Rate of the model.

---

> ### Author Response · Authors · 2025-11-28
> **Rebuttal by Authors: W4, Q1, Q2**
>
> ## W4: Readability and Organization
>
> **Reviewer'Comment:**  While clear overall, the draft is verbose in places and could improve figure readability and section organization. E.g. Figure 1 contained red wavy dash, need fix.
>
> **Our Response:**
>
> We acknowledge the identified error in Figure 1 and have corrected it. Additionally, we will improve the organization of sections in the final version to enhance clarity.
>
>
> ## Q1: Fairness of Task Difficulty
>
> **Reviewer'Comment:**  How was task difficulty balanced between GUI-only and MCP-enhanced versions to ensure fairness?
>
> **Our Response:**
>
> 1. **Unified Task Framework**.
> We did not construct two separate versions for a direct difficulty comparison. Instead, our benchmark provides a **unified task framework**. In line 343, *"With the introduction of MCP tools in our OSWorld‑MCP, the LMM can autonomously decide whether to invoke any MCP Tool or perform a GUI operation at each action step. In detail, we employ the 158 high‑quality MCP tools curated in Section 3.2."*  The primary goal is to jointly assess **an agent’s tool invocation capability, GUI operation skills, and decision-making competence** within an integrated environment.
>
> 2. **Addressing Fairness Limitations in Original OSWorld**.
> The original OSWorld did not consider **the inclusion of MCP tools**, while MCP tools are becoming increasingly prevalent. Under the original OSWorld setting, directly comparing agents equipped with built-in APIs to those limited to GUI operations is **inherently unfair**. Therefore, it is necessary to introduce OSWorld-MCP, a **comprehensive and fair** benchmark that integrates the evaluation of tool invocation capabilities, GUI skills, and decision-making abilities.
>
> 3. **Task Categorization**.
> In our benchmark, there is **no distinction between GUI-only and MCP-enhanced versions**. Instead, we define Tool-Beneficial Tasks as tasks for which at least one available tool can improve efficiency, and Non-Tool-Beneficial Tasks as tasks for which no available tool provides efficiency gains. This classification enables us to evaluate the efficiency with which agents complete tasks and assess whether they are able to employ the most efficient approach. According to Tables 1 and 3, even the most advanced models achieve a tool invocation rate of only 36.3%, indicating substantial room for improvement in tool invocation capability in future models.
>
>
> ## Q2: Tool Invocation Failures Classification
>
> **Reviewer'Comment:**  Could tool invocation failures be categorized (e.g., recognition vs. reasoning vs. execution errors)?
>
> **Our Response:**
>
> To analyze the causes of agents’ tool invocation failures, we randomly selected **524 tasks** with problematic tool usage for detailed examination. We categorized these failure cases into three groups: decision-making errors, MCP tool invocation errors, and MCP tool parameter errors. The statistical results are presented in the following table.
>
> | Model            | Reasoning failure | MCP tool invocation error | MCP tool parameter error | Overall |
> |------------------|-------------------|---------------------------|--------------------------|---------|
> | Qwen2.5-VL       | 151 (68.3%)       | 28 (12.7%)                  | 42 (19.0%)               | 221     |
> | OpenAI o3        | 158 (89.3%)       | 7 (4.0%)                   | 12 (6.8%)                 | 177     |
> | Claude 4 Sonnet  | 119 (94.4%)       | 4 (3.2%)                   | 3 (2.4%)                  | 126     |
>
> **Analysis**:
>
> We observed that as **agent capabilities improve**, the proportions of invocation errors and parameter errors gradually decrease, whereas the proportion of **Decision-making errors increases** steadily.
>
>
> ## Reference
> Hou X, Zhao Y, Wang S, et al. Model context protocol (mcp): Landscape, security threats, and future research directions[J]. arXiv preprint arXiv:2503.23278, 2025.
>
> Chhetri G, Somvanshi S, Islam M M, et al. Model Context Protocols in Adaptive Transport Systems: A Survey[J]. arXiv preprint arXiv:2508.19239, 2025.
>
> Hasan M M, Li H, Fallahzadeh E, et al. Model context protocol (mcp) at first glance: Studying the security and maintainability of mcp servers[J]. arXiv preprint arXiv:2506.13538, 2025.
>
> Gonzalez-Pumariega G, Tu V, Lee C L, et al. The Unreasonable Effectiveness of Scaling Agents for Computer Use[J]. arXiv preprint arXiv:2510.02250, 2025.

---

### Official Review · Reviewer_cXcb · 2025-11-01

**Soundness:** 3
**Presentation:** 3
**Contribution:** 2
**Rating:** 6
**Confidence:** 4

**Summary:**

This paper introduces OSWorld-MCP, a new benchmark for evaluating AI agents' ability to use both GUI and external tools via the Model Context Protocol (MCP). The authors developed a set of 158 high-quality tools and found that using them generally improves agent performance on various computer tasks. However, the results also show that even the most advanced models still struggle with effectively invoking these tools, indicating significant room for improvement.

**Strengths:**

1. **Creation of a High-Quality and Rigorously Validated Toolset**: The paper's primary contribution is the construction of a novel benchmark centered around a "curated collection of 158 high-quality tools". The reliability of this toolset is strongly supported by a meticulous creation process, which combines a "novel automated code-generation pipeline" with "rigorous manual validation". This validation was not superficial; each tool was independently assessed by at least two experienced reviewers to ensure its "correct functionality, practical applicability, and versatility", establishing a robust and trustworthy foundation for future agent evaluation.
2. **Introduction of More Granular and Insightful Evaluation Metrics**: Moving beyond simple task accuracy, the paper introduces two new metrics—Tool Invocation Rate (TIR) and Average Completion Steps (ACS)—to enable a "more nuanced evaluation" of agent capabilities. TIR offers crucial insights into an agent's "tool utilization propensity," while ACS directly "quantifies task completion efficiency".
3. **A Unified Benchmark Bridging GUI and Tool-Use (MCP)**: A significant strength is the benchmark's design, which integrates GUI operations with MCP tool invocation in a single, dynamic environment. This approach directly addresses the "gap between pure-GUI and text-based tool-use evaluations", creating the first "comprehensive and fair benchmark" of its kind. By forcing agents to autonomously choose between GUI actions and MCP tools at each step, OSWorld-MCP provides a "holistic and realistic assessment" of an agent's hybrid decision-making skills in complex, real-world computer tasks.

**Weaknesses:**

1. **Incremental Novelty Built on an Existing Framework**: While the work is a significant engineering effort, its core contribution is presented as an extension to the pre-existing OSWorld benchmark. The paper states it is "Built upon a widely used... environment OSWorld" [cite: 077-078]. Consequently, the foundational novelty could be viewed as somewhat limited, as it enhances an established platform rather than introducing a completely new paradigm for agent evaluation.
2. **Insufficient Depth in Analyzing the Hybrid GUI-MCP Interaction**: The paper's central premise is that evaluating the choice between GUI and MCP is more realistic. However, the analysis could delve deeper into the nuances of this hybrid decision-making. While the results in Table 1 show that adding MCP tools improves outcomes, the paper does not offer a detailed qualitative analysis of the agents' decision-making process itself. For example, it lacks a thorough exploration of common failure modes, such as when an agent illogically prefers a complex GUI sequence over a simple tool, which would offer richer insights into the current reasoning limitations of models.
3. **Current Limitations in Application and Task Diversity**: The benchmark's scope is currently constrained, as the authors acknowledge that "MCP servers were not developed for GIMP and Thunderbird". This gap limits the benchmark's ability to assess agent performance across a truly comprehensive range of real-world activities and narrows the generalizability of its conclusions.

**Questions:**

1. Could you please provide more details on the Retrieval-Augmented Generation (RAG) system? Specifically, what retriever model was used, and how was the query formed at each step to filter the 158 tools?
2. What was the exact prompt template used to integrate the visual information (screenshot), the task instruction, and the descriptions of the RAG-filtered MCP tools for the LMM to make a decision at each step?

---

> ### Author Response · Authors · 2025-11-28
> **Rebuttal by Authors: W1**
>
> We sincerely thank the reviewer for the thoughtful feedback and for recognizing the potential of our work. We address the main points as follows.
>
> ## W1: Incremental Novelty Built on an Existing Framework
>
> **Reviewer'Comment:**  Incremental Novelty Built on an Existing Framework: While the work is a significant engineering effort, its core contribution is presented as an extension to the pre-existing OSWorld benchmark. The paper states it is "Built upon a widely used... environment OSWorld" [cite: 077-078]. Consequently, the foundational novelty could be viewed as somewhat limited, as it enhances an established platform rather than introducing a completely new paradigm for agent evaluation.
>
> **Our Response:**
>
> We present **the first benchmark** to **fairly and comprehensively** evaluate **the tool invocation capability, GUI operation capability, and decision-making ability** of computer-use agents. Reviewer 8XpA commented that our methodology is solid and the significance is high. Reviewer DEYi stated that we proposed a novel and comprehensive benchmark framework and that the core idea of combining MCP tools with a GUI agent is novel, and further noted that our experimental design is very detailed and that we did a great job in discussing the results.
>
> Our key innovations include:
> - **MCP tool invocation evaluation:** First benchmark to assess agents’ ability to use MCP tools within GUI tasks, a skill increasingly critical for AI-native applications.
> - **Fair, integrated benchmarking:** Jointly evaluates tool use, GUI operation skills, and decision-making, addressing inequities in comparing GUI-only and API-equipped agents.
> - **Automated tool generation pipeline:** Combines automated code generation and manual curation to create high-quality MCP tools, broadly applicable for future benchmarks.
> - **New metrics:** Tool Invocation Rate (TIR) for measuring tool use propensity and Average Completion Steps (ACS) for task efficiency.

---

> ### Author Response · Authors · 2025-11-28
> **Rebuttal by Authors: W2**
>
> ## W2: Insufficient Depth in Analyzing the Hybrid GUI-MCP Interaction
>
> **Reviewer'Comment:**  Insufficient Depth in Analyzing the Hybrid GUI-MCP Interaction: The paper's central premise is that evaluating the choice between GUI and MCP is more realistic. However, the analysis could delve deeper into the nuances of this hybrid decision-making. While the results in Table 1 show that adding MCP tools improves outcomes, the paper does not offer a detailed qualitative analysis of the agents' decision-making process itself. For example, it lacks a thorough exploration of common failure modes, such as when an agent illogically prefers a complex GUI sequence over a simple tool, which would offer richer insights into the current reasoning limitations of models.
>
> **Our Response:**
>
> Thank you for your insightful question.
>
> First, as described in Section 3.4 and Table 1, our TIR metric can reflect the model’s ability to select the correct interaction mode (GUI or MCP) and successfully complete the task. Specifically, it distinguishes between Tool-Beneficial and Non-Tool-Beneficial tasks, measuring whether the model invoked a tool when appropriate and avoided it when unnecessary, thus directly indicating the correctness of its choice strategy in hybrid scenarios.
>
> While TIR provides an aggregate measure, as suggested by the reviewer, we performed a detailed error analysis of **250 Tool‑Beneficial tasks that require invoking tools** for Qwen2.5-VL, OpenAI o3, Claude 4 Sonnet and identified three main decision-making errors.
>
> (1) **GUI‑overreliance errors:** relying solely on GUI actions and failing to recognize tasks that require tools.
>
> (2) **Tool‑misuse errors:** incorrect tool type or parameters, indicating weak invocation reasoning.
>
> (3) **GUI‑tool integration errors:** invoking the correct MCP tool, but failed to integrate the tool with GUI operations, showing flaws in task decomposition.
>
> The distribution of these errors is as follows:
>
> | Model           | GUI‑overreliance errors | Tool‑misuse errors | GUI‑tool integration errors | Total |
> |-----------------|-----------------------|--------------------|------------------------------|-------|
> | Qwen2.5‑VL      | 69             | 67                 | 81                           | 217   |
> | OpenAI o3       | 32             | 63                 | 67                           | 162   |
> | Claude 4 Sonnet | 21             | 57                 | 33                           | 111   |
>
> **Analysis:**
>
> (1) Across models, these three types of errors in hybrid GUI-MCP interaction are considerable.
> Even for the most advanced models (Claude 4 Sonnet), the error proportion remains as high as 44.4% (111/250), indicating substantial room for improvement in dynamic GUI‑MCP decision‑making capability.
>
> (2) **Tool‑misuse** errors occur more frequently, highlighting shortages in the reasoning and mapping from task requirements to the correct tool invocation. This suggests that **improving tool selection logic should be a key priority**.
>
> (3) **GUI‑overreliance** and **GUI‑tool integration errors** decrease with more capable models, but weaker models (Qwen2.5-VL) still struggle with accurate “routing” decisions (whether to invoke a tool or to proceed via GUI). It suggests that an explicit routing mechanism during inference may be useful.
>
> (4) **GUI‑tool integration errors** often arise in tasks with **more subtasks**. Even advanced models sometimes omit steps or mis-sequence actions. Examples below illustrate this type of error:
> - Example: *Instruction: Give the slide a purple background color. Add the title to note.*
>   Claude 4 Sonnet solved this task by correctly setting the slide’s background color using the MCP tool `set_slide_background` together with GUI operations, but it omitted the “Add the title to note” instruction and ended the task prematurely, resulting in incomplete task execution.
> - Example: *Instruction: Append one entry of AI researcher Yann LeCun from Google Scholar into an existing table researchers.xlsx.*
> Claude 4 Sonnet used the MCP tools `env_info`, `chrome_open_tabs_setup`, `set_cell_value` and `save`. However, it fabricated a PDF link of the paper instead of obtaining the genuine one through appropriate GUI operations (e.g., drag to select and ctrl+c to copy).
>
> These examples indicate that overall task comprehension remains a challenge for agents. We recommend two directions for improvement:
> - Introduce richer world knowledge to enhance understanding of complex tasks.
> - Improve global planning capability to avoid limitations caused by restricted global context.

---

> ### Author Response · Authors · 2025-11-28
> **Rebuttal by Authors: W3, Q1, Q2**
>
> ## W3: Current Limitations in Application and Task Diversity
>
> **Reviewer'Comment:**  Current Limitations in Application and Task Diversity: The benchmark's scope is currently constrained, as the authors acknowledge that "MCP servers were not developed for GIMP and Thunderbird". This gap limits the benchmark's ability to assess agent performance across a truly comprehensive range of real-world activities and narrows the generalizability of its conclusions.
>
> **Our Response:**
>
> 1. **Current Coverage**
>
> Our OSWorld-MCP currently covers **most commonly used applications** in OSWorld and maintains a set of 158 high-quality MCP tools. OSWorld-MCP includes frequently encountered real-world scenarios such as web browsing (Chrome), media playback (VLC), programming in an integrated development environment (VSCode), handling spreadsheets, documents, and presentations (LibreOffice Calc, Writer, and Impress), as well as other basic operating system applications (OS). Example tasks include:
> - LibreOffice Impress: Move the table on Page 3 to the bottom of the slide.
> - VS Code: Set the current user's line length for code wrapping to 50 characters in VS Code.
> - Chrome: Find the score record for the 2019 Super Bowl on the NFL website.
>
> 2. **Reasons for Exclusion**
>
> We intentionally did not include GIMP and Thunderbird in this release because the available MCP tools for these applications are **limited in number and lack generality**.
> - GIMP’s interface is developed using GTK, while Thunderbird employs the XML User Interface Language, making the development of GUI-layer tools comparatively complex. The tools generated by our automated tool generation process are **highly customized for specific tasks** and were removed during manual review to maintain the quality of the benchmark. For example, replying to an email with quoted original text without adding indentation or special characters.
> - Publicly available MCP tools for GIMP and Thunderbird (LibreEarth et al., 2025) are often **overly specialized** (for instance, “Connect GIMP with state-of-the-art AI models for enhanced image processing capabilities”) or incompatible with our OSWorld configuration (e.g., **version mismatches**, such as GIMP 2.10 versus MCP servers built for version 3.0 and above).
>
>
> ## Q1: Details on RAG
>
> **Reviewer'Comment:**  Could you please provide more details on the Retrieval-Augmented Generation (RAG) system? Specifically, what retriever model was used, and how was the query formed at each step to filter the 158 tools?
>
> **Our Response:**
>
> Our RAG module adopts a lightweight yet effective pipeline:
>
> Step 1: Detect the software currently in use by monitoring active application context within the OSWorld environment.
>
> Step 2: Filter MCP tools corresponding to that software in our curated tool set. If no software is currently open, OS-related MCP tools are passed in instead.
>
> Step 3: Within the reduced candidate set, retrieve relevant tools using simple keyword-based matching on tool names and descriptions. This avoids overfitting to training data and keeps retrieval interpretable.
>
> The results show that this simple filtering combined with keyword-based retrieval is **sufficient to reliably return the correct tools**. Future iterations of the evaluation framework can further refine the RAG pipeline to reduce the number of recalled MCP tools.
>
> ## Q2: Prompt Template
>
> **Reviewer'Comment:**  What was the exact prompt template used to integrate the visual information (screenshot), the task instruction, and the descriptions of the RAG-filtered MCP tools for the LMM to make a decision at each step?
>
> **Our Response:**
>
> As illustrated in the **Appendix B**, our prompt template integrates visual information (screenshot), task instructions, and descriptions of the MCP tools filtered by retrieval-augmented generation (RAG). This design provides the LMM with all necessary contextual information to make decisions at each step between MCP-based actions and GUI operations.
>
> We thank the reviewer again for the constructive comments, which have helped us strengthen the clarity and emphasis on our contributions.
>
> **References**
> Deng M, Dong G, Dou Z. ToolScope: An Agentic Framework for Vision-Guided and Long-Horizon Tool Use[J]. arXiv preprint arXiv:2510.27363, 2025.
>
> LibreEarth. gimp-mcp[EB/OL]. (2025-11-26). https://github.com/libreearth/gimp-mcp.

---

### Official Review · Reviewer_8XpA · 2025-11-02

**Soundness:** 3
**Presentation:** 3
**Contribution:** 3
**Rating:** 8
**Confidence:** 3

**Summary:**

The paper introduces OSWorld-MCP, a new benchmark built on OSWorld for evaluating multimodal computer-use agents that perform both GUI operations and Model Context Protocol (MCP) tool invocations. It extends OSWorld by adding 158 validated MCP tools across seven common applications and supplements existing metrics with two new ones—Tool Invocation Rate (TIR) and Average Completion Steps (ACS)—to measure tool-use propensity and efficiency. An automated pipeline using OpenAI o3 is proposed to generate and curate high-quality tools. Experiments on several state-of-the-art multimodal models demonstrate that MCP tools substantially improve task performance, though multi-tool composition remains challenging. Overall, OSWorld-MCP offers a fair and realistic framework for benchmarking tool-augmented computer-use agents.

**Strengths:**

The paper is original in extending OSWorld to evaluate MCP-based tool invocation, a capability overlooked in prior benchmarks. Its methodology is solid, combining automated tool generation with manual validation and extensive experiments on several state-of-the-art models. The presentation is clear and well-structured. The significance is high—OSWorld-MCP offers a realistic, open benchmark that will benefit research on tool-using multimodal agents. Its usefulness will further grow as more GUI applications adopt MCP-compatible tools.

**Weaknesses:**

The benchmark’s scope is somewhat limited—its 158 tools cover mainly seven desktop apps, leaving out broader web or cross-platform settings. The tool-generation pipeline lacks quantitative evaluation of success and failure cases. Some tasks remain similar to the original OSWorld, reducing novelty in task design. While TIR and ACS are useful, their interpretation could be clearer. Finally, the benchmark’s long-term value depends on wider MCP adoption across GUI applications.

**Questions:**

1.Tool Description Impact: How does the phrasing or length of tool descriptions influence model performance? Have the authors observed any sensitivity to wording or order beyond the shuffle experiment?

2.Failure Case Analysis: Can the authors provide an analysis of common failure cases—specifically, whether models fail by selecting the wrong tool or by invoking the correct tool with incorrect arguments?

3.Metric Interpretation: How should Tool Invocation Rate (TIR) be interpreted when models invoke tools unnecessarily or inefficiently—does a higher TIR always indicate better decision-making quality?

---

> ### Author Response · Authors · 2025-11-28
> **Rebuttal by Authors: W1, W2**
>
> We sincerely thank you for your positive assessment of our work. We highly appreciate the valuable comments and questions you have raised. Below, we provide detailed clarifications and additional information addressing the identified weaknesses and questions.
>
> ## W1: The Benchmark’s Scope is Somewhat Limited
>
> **Reviewer'Comment:** The benchmark’s scope is somewhat limited—its 158 tools cover mainly seven desktop apps, leaving out broader web or cross-platform settings.
>
> **Our Response:**
>
> ### 1. Clarifying the Scope and Motivation
>
> As stated in the introduction, our OSWorld-MCP follows the scope of OSWorld, which is the most widely adopted and most reproducible real-world GUI benchmark. By retaining this setup, OSWorld-MCP is **fully capable of evaluating computer-use agents’ MCP tool invocation abilities** in realistic scenarios.
>
> To ensure **fairness** in comparison and **efficiency** in tool usage within OSWorld-MCP, we extend OSWorld by adding 158 high-quality MCP tools. Our automated tool generation framework yields a substantial collection of readily usable tools. These tools cover all evaluation scenarios in OSWorld, ensuring both consistency and fairness.
>
> ### 2. OSWorld-MCP’s scope goes beyond purely desktop apps
> Our OSWorld-MCP already includes web-based and cross-application. In particular:
>
> **Web-based tasks**: 46 tasks involve web interactions through Chrome, such as:
> - Chrome/7f52cab9-535c-4835-ac8c-391ee64dc930："Create a list of drip coffee makers that are on sale and within $25-60 and have a black finish."
> - Chrome/82279c77-8fc6-46f6-9622-3ba96f61b477："Find electric cars with a maximum price of $50,000 within 50 miles of 10001."
>
> Web-related MCP tools include:
> - `chrome_open_tabs_setup`: Opens the entered URL
> - `bring back last tab`: Restores the last-closed tab in the browser
> - `bookmark page`: Bookmarks the current page in the browser
>
> **Cross-application tasks**. 101 tasks require interaction across multiple apps, such as:
>
> - multi_apps/5990457f-2adb-467b-a4af-5c857c92d762：Append one entry of AI researcher Yann LeCun from Google Scholar into an existing table researchers.xlsx.
> - multi_apps/00fa164e-2612-4439-992e-157d019a8436：I need to include the experiment results from \"~/Documents/awesome-desktop/expe-results.xlsx\" into the currently writing report. Specifically, extract the results of GPT-4 and insert a table into the \"Main Results\" section of my report.
>
> Moreover, our automated tool generation process facilitates the production of additional cross-platform tools.
>
>
> ## W2: The Quantitative Evaluation of The Tool-Generation Pipeline
>
> **Reviewer'Comment:**  The tool-generation pipeline lacks quantitative evaluation of success and failure cases.
>
> **Our Response:**
>
> Thank you for your suggestion. We have added a quantitative evaluation of the automated tool generation pipeline. Specifically, in our pipeline, we automatically generate 72 usable tools from 113 tasks selected for automated code generation, resulting in a tool conversion rate of **63.7%.** The task-to-tool conversion for each application is shown in the table below:
>
> | App                      | Chrome | LibreOffice Calc | LibreOffice Impress | LibreOffice Writer | OS | Thunderbird | VLC | VS Code | GIMP |
> |--------------------------|--------|------------------|---------------------|--------------------|----|-------------|-----|---------|------|
> | Number of tasks for code generation | 10 | 18 | 15 | 13 | 26 | 3 | 11 | 14 | 3 |
> | Number of converted tools            | 6  | 15 | 5  | 5  | 21 | 2 | 6  | 10 | 2 |
> | tool conversion rate (%)            | 60  | 83 | 33  | 38  | 81 | 67 | 55  | 71 | 67 |
>
> As shown in the table, **the conversion rate for LibreOffice Impress is the lowest, while Thunderbird and GIMP have the fewest tools obtained through conversion**.
> - LibreOffice Impress is a presentation software with **many available APIs**, yet the tasks can be **complex**. For example, "I want to do something like the following on Page 3 of the current LibreOffice Impress file: make the 'continuation of the sub topics' indented the same as 'main idea' and 'sub topics' without a bullet in front of it. Could you help me with it?" During task execution, it is difficult to generate non-custom tools that can distinguish between elements of the same type, such as using a tool to determine the indentation of “main idea,” which results in a low tool conversion rate.
> - In contrast, GIMP and Thunderbird have very **limited available tools**. For instance, in the task "Could you help me open up the profile management tabpage in Thunderbird? I want the profile management tabpage inside the Thunderbird app, but not the profile chooser dialog during app launch," there is no available API, and using the GUI offers a more convenient solution.

---

> ### Author Response · Authors · 2025-11-28
> **Rebuttal by Authors: W3, W4**
>
> ## W3: The Task Design Novelty
>
> **Reviewer'Comment:**  Some tasks remain similar to the original OSWorld, reducing novelty in task design.
>
> **Our Response:**
>
> 1. **Reason for Reusing OSWorld Tasks**
>
>   - OSWorld is the **most widely used and most reproducible** benchmark in the GUI domain (Claude Opus 4.5, Qwen3‑VL, and OpenAI CUA are all evaluated using it). Using the same task set ensures **direct comparability** with prior work and enables **fair evaluation** of agents under identical scenarios.
>
>   - The task design in OSWorld covers **a broad range of real-world scenarios**, and each task is accompanied by a well-defined evaluation protocol. Therefore, adopting the same task setting is **sufficient** for OSWorld-MCP to **assess the capabilities** of computer-use agents **in realistic environments**. OSWorld-MCP encompasses commonly used real-world scenarios such as web browsing (Chrome), media playback (VLC), programming in an integrated development environment (VSCode), handling spreadsheets, documents, and presentations (LibreOffice Calc, Writer, and Impress), as well as other basic operating system applications (OS). It also involves a broad set of general MCP tools, for example:
>     - LibreOffice Calc set_cell_value: set a value to a specific cell in the active worksheet.
>     - LibreOffice Impress write_text: write text to a specific textbox on a slide.
>     - LibreOffice Writer save: save the current document to its current location.
>
>   This enables OSWorld-MCP to comprehensively evaluate the MCP tool usage capabilities of computer-use agents in real-world scenarios.
>
> 2. **Novelty Lies in Tool-based Evaluation Setting**
>
>   Although the task set remains unchanged, our OSWorld-MCP shifts the focus from purely GUI-based operations to tool usage and decision-making in a hybrid environment. Two reviewers have also acknowledged our innovations. Our key innovations are as follows:
>   - **MCP tool invocation evaluation:** First benchmark to assess agents’ ability to invoke MCP tools within GUI tasks. MCP usage can be more efficient and robust than GUI operations and is increasingly fundamental to AI-native applications  (Hou et al., 2025; Hasan et al., 2025; Chhetri et al., 2025).
>   - **Fair, integrated evaluation:** Existing GUI benchmarks ignore MCP tools, making comparisons between GUI-only agents and API-equipped agents inequitable. OSWorld-MCP jointly evaluates tool use, GUI operation, and decision-making in one framework (Xie et al., 2024; Bonatti et al., 2024; Kuntz et al., 2025).
>   - **Automated tool generation pipeline:** Novel combination of automated code generation and manual curation to create high-quality MCP tools, broadly applicable in future work.
>   - **New metrics** – Tool Invocation Rate (TIR) to quantify tool utilization propensity and Average Completion Steps (ACS) to measure task completion efficiency.
>
> ## W4: Interpretation of Metrics TIR and ACS
>
> **Reviewer'Comment:**  While TIR and ACS are useful, their interpretation could be clearer.
>
> **Our Response:**
>
> We appreciate your suggestion and will incorporate the detailed descriptions of Tool Invocation Rate (TIR) and Average Completion Steps (ACS) into the final version.
>
> *Tool Invocation Rate (TIR):* TIR effectively measures a multimodal agent's **ability to utilize tools** in relevant scenarios. As described in Section 4.2, reviewers categorize tasks into Tool-Beneficial or Non-Tool-Beneficial. Let $N_t$ be the number of Tool-Beneficial tasks and $n_t$ the number successfully completed with tool invocation. Let $N_g$ denote Non-Tool-Beneficial tasks and $n_g$ the number successfully completed without tool invocation. We define
> $$
> \text{TIR} = \frac{n_t + n_g}{N_t + N_g}
> $$
>
> For Tool-Beneficial subsets $(N_g=0)$, $\text{TIR} = n_t / N_t$, measuring the proportion of Tool-Beneficial Tasks where the agent correctly invoked a tool and succeeded. For Non-Tool-Beneficial subsets $(N_t=0)$, $\text{TIR} = n_g / N_g$, measuring the proportion of Non-Tool-Beneficial Tasks where the agent correctly avoided tool invocation and succeeded.
>
>
> *Average Completion Steps (ACS):* This metric measures **the average number of steps** an agent takes to complete a task. ACS reflects **decision-making efficiency**: the more accurate the decisions, the higher the rate of correct tool usage, and the more frequently the agent selects efficient tools, the lower ACS will be. For example, choosing an MCP Tool over a GUI operation when both can accomplish the task, the ACS will be lower.
>
> For $N$ tasks, if the number of execution steps for task $i$ is $S_i$, the Average Completion Steps is computed as:
> $$
> \text{ACS} = \frac{\sum_{i=1}^{N} S_i}{N}
> $$

---

> ### Author Response · Authors · 2025-11-28
> **Rebuttal by Authors: W5, Q1**
>
> ## W5: Benchmark’s Long-term Value
>
> **Reviewer'Comment:**  Finally, the benchmark’s long-term value depends on wider MCP adoption across GUI applications.
>
> **Our Response:**
>
> Thank you for this very professional and forward-looking question. In fact, current trends indicate rapid growth in MCP servers and tools (Gonzalez-Pumariega et al., 2025; Song et al., 2025; Abhyankar et al., 2025). The GUI and MCP can work together synergistically to solve tasks **more efficiently**. Computer-use agents that can **flexibly choose** between GUI operations, MCP tools, and applications are able to complete tasks **more robustly**.
>
> Besides, as MCP adoption grows, our platform-agnostic tool generation pipeline can easily incorporate new MCP tools and applications, ensuring the benchmark **remains relevant and scalable in the long term**.
>
>
> ## Q1: Impact of Tool Description Phrasing, Length, and Order
>
> **Reviewer'Comment:**
> Tool Description Impact: How does the phrasing or length of tool descriptions influence model performance? Have the authors observed any sensitivity to wording or order beyond the shuffle experiment?
>
> **Our Response:**
>
> The **length and phrasing** of tool descriptions, as well as **the ordering of tools**, have **very little impact on the evaluation results** in OSWorld-MCP. Overall, the OSWorld-MCP benchmark demonstrates **strong robustness** in its assessment.
>
> To validate this, we conducted additional ablation experiments:
>
> - Description phrasing styles: Using OpenAI o3, we rewrote each tool and parameter description into four styles: **concise, standard, detailed, and ornate versus plain**, and preserve semantic meaning and keep lengths comparable.
>
> - Tool ordering: MCP tools were sorted in three ways—parameter-based ascending order, random order, and pure lexicographical order.
>
> - Description lengths: Created short, medium, and detailed versions of each description to test sensitivity.
>
> (1) Phrasing style results:
>
> | Model             | Description Style | Avg Tokens | Acc (%) | TIR (%) | ACS  |
> |-------------------|-------------------|------------|---------|---------|------|
> | Gemini-2.5 Pro    | Concise           | 17.8        | 20.3    | 17.4    | 11.2 |
> | Gemini-2.5 Pro    | Original          | 17.5        | 20.5    | 16.8     | 11.4 |
> | Gemini-2.5 Pro    | Ornate             | 18.6       | 20.1    | 16.3    | 11.5 |
> | Claude 4 Sonnet   | Concise           | 17.8        | 35.0    | 29.6    | 10.5 |
> | Claude 4 Sonnet   | Original          | 17.5        | 35.3  | 30.0    |  10.4 |
> | Claude 4 Sonnet   | Ornate             | 18.6       | 35.5    | 31.0    | 10.4 |
>
>
> (2) Description length results:
>
> | Model             | Length Setting    | Avg Tokens | Acc (%) | TIR (%) | ACS  |
> |-------------------|-------------------|------------|---------|---------|------|
> | Gemini-2.5 Pro    | Short             | 17.5         | 20.5    | 16.8    | 11.4 |
> | Gemini-2.5 Pro    | Medium            | 23.2         | 20.9    | 16.1    | 11.6 |
> | Gemini-2.5 Pro    | Detailed          | 46.2         | 21.3    | 19.9    | 11.4 |
> | Claude 4 Sonnet   | Short             | 17.5         | 35.3  | 30.0      | 10.4 |
> | Claude 4 Sonnet   | Medium            | 23.2         | 35.5    | 29.1    | 10.4 |
> | Claude 4 Sonnet   | Detailed          | 46.2         | 34.8    | 29.6    | 10.8 |
>
> (3) Tool order results:
>
> | Model             | Tool Order        | Acc (%) | TIR (%) | ACS  |
> |-------------------|-------------------|---------|---------|------|
> | Gemini-2.5 Pro    | Dictionary Order  | 20.5    | 16.8     | 11.4 |
> | Gemini-2.5 Pro    | Random Order      | 22.7    | 18.6    | 11.5 |
> | Gemini-2.5 Pro    | Parameter Order   | 21.4    | 16.1    | 10.7 |
> | Claude 4 Sonnet   | Dictionary Order  | 35.3    | 30.0    | 10.4 |
> | Claude 4 Sonnet   | Random Order      | 35.1    | 29.4    | 10.3 |
> | Claude 4 Sonnet   | Parameter Order   | 35.8    | 29.6    | 10.8 |
>
> ### Analysis:
> Across all ablations, differences in Acc, TIR, and ACS are within ±1–2%, showing minimal sensitivity to phrasing, length, or ordering for both models. We attribute this robustness to:
>
> - Semantic preservation: Modifications did not change the functional meaning of tools, enabling models to interpret and act consistently.
> - Task nature: Success depends more on high-level decision-making than minor textual variations.
>
> This robustness is important for benchmark reliability—OSWorld-MCP yields stable and comparable results across varying description styles, lengths, and orders, ensuring fairness and reproducibility in long-term evaluations.

---

> ### Author Response · Authors · 2025-11-28
> **Rebuttal by Authors: Q2, Q3**
>
> ## Q2: Failure Case Analysis
>
> **Reviewer'Comment:**  Failure Case Analysis: Can the authors provide an analysis of common failure cases—specifically, whether models fail by selecting the wrong tool or by invoking the correct tool with incorrect arguments?
>
> **Our Response:**
>
> We analyzed 785 failure cases in a single round of evaluation from multiple agents in OSWorld-MCP, and categorized them into four types: **Reasoning failures**, **Grounding failures**, **MCP tool invocation errors** and **MCP tool parameter errors**, as described in Appendix A.2.
>
> The distribution across models is shown below:
>
>
> | Model            | Reasoning failure | Grounding failure | MCP tool invocation error | MCP tool parameter error | Overall |
> |------------------|-----------------------------|---------------------------|--------------------------------|------------------------------|------------------------------|
> | Qwen2.5-VL       | 227 (73.5%)                 |  12 (3.9%)                   | 28 (9.1%)                       | 42 (13.6%)                     | 309 |
> | OpenAI o3        | 201 (74.2%)                   | 51 (18.8%)                  | 7 (2.6%)                             | 12 (44.3%)                           | 271 |
> | Claude 4 Sonnet  | 197 (96.1%)                   | 1 (0.5%)                    | 4 (2.0%)                         | 3 (1.5%)                       | 205 |
>
> **Analysis**:
>
> Reasoning failures are the most common type of error among all agents. For powerful models such as Claude 4 Sonnet, although low-level GUI and tool errors are rare, reasoning failures still persist. As agent capabilities improve, GUI and tool parameter errors tend to decrease, yet in a hybrid GUI and tool environment, high-level decision-making errors remain the most challenging issue to address. This provides guidance for **future work**, indicating that for computer-use agents, **further improvement in reasoning ability is essential**.
>
>
>
> ## Q3: Combined Utility of TIR and ACS
>
> **Reviewer'Comment:**  Metric Interpretation: How should Tool Invocation Rate (TIR) be interpreted when models invoke tools unnecessarily or inefficiently—does a higher TIR always indicate better decision-making quality?
>
> **Our Response:**
>
> In our view, the combination of Tool Invocation Rate (TIR) and Average Completion Steps (ACS) provides informative signals regarding unnecessary or inefficient tool usage by agents. ACS measures the average number of steps required to complete a task, and a higher ACS often indicates cases where an agent invokes tools unnecessarily or in a suboptimal manner. When considered together, **TIR and ACS reflect the efficiency of tool utilization**: a higher TIR accompanied by a lower ACS suggests that the agent is effectively and correctly invoking tools to accomplish tasks more efficiently.
>
> ## Reference
> Hou X, Zhao Y, Wang S, et al. Model context protocol (mcp): Landscape, security threats, and future research directions[J]. arXiv preprint arXiv:2503.23278, 2025.
>
> Chhetri G, Somvanshi S, Islam M M, et al. Model Context Protocols in Adaptive Transport Systems: A Survey[J]. arXiv preprint arXiv:2508.19239, 2025.
>
> Hasan M M, Li H, Fallahzadeh E, et al. Model context protocol (mcp) at first glance: Studying the security and maintainability of mcp servers[J]. arXiv preprint arXiv:2506.13538, 2025.
>
> Gonzalez-Pumariega G, Tu V, Lee C L, et al. The Unreasonable Effectiveness of Scaling Agents for Computer Use[J]. arXiv preprint arXiv:2510.02250, 2025.

---

### Author Response · Authors · 2025-11-28
**Summary of updates in the revised manuscript**

We thank all the reviewers and the AC for their valuable feedback and contributions during the discussion period. Below we summarize the updates made in the revised manuscript, which we have just submitted.

**Additional Results and Analysis:**
- Results averaged over three runs: see Table 1, Table 2, and Figure 5.
- Additional bad case analyses:
  - Reasoning failure (Appendix A.2.1)
  - Grounding failure (Appendix A.2.2)
  - MCP tool invocation error (Appendix A.2.3)
  - MCP tool parameter error (Appendix A.2.4)
- Variance measures and confidence intervals (Appendix A.3 and Table 3)
- Added prompt template description (Appendix B).

**Writing Improvements:**
- Merged experimental results from Table 3 into Table 1.
- Expanded Section 3.4 with explanations of TIR and ACS.

---

### Meta-Review · Area_Chair_nrY6 · 2026-01-12

**Summary:**

This paper introduces OSWorld-MCP, a benchmark extending OSWorld to evaluate computer-use agents that must dynamically choose between GUI actions and MCP tool invocations. Reviewers broadly agreed that the benchmark fills an important gap left by GUI-only or API-only evaluations, and that the automated tool-generation pipeline combined with rigorous manual validation (158 tools across common applications) is a substantial engineering contribution. Experiments across multiple state-of-the-art agents show consistent performance gains from MCP tools while revealing that correct tool usage remains rare, highlighting a meaningful and non-trivial bottleneck. Overall, the submission is viewed as a timely and useful benchmark contribution to agentic and multimodal AI evaluation.

**Reviewer Concerns:**

Most substantive reviewer concerns were addressed in the rebuttal. In particular, the authors clarified the benchmark scope and task coverage, provided quantitative evaluation of the tool-generation pipeline, added statistical robustness analyses (multiple runs, confidence intervals, and significance tests), and substantially expanded failure-mode analyses for hybrid GUI–MCP decision-making. Reviewers’ questions about metric interpretation (TIR, ACS), sensitivity to tool description phrasing/order, and RAG-based tool filtering were answered with additional experiments and concrete ablations.

The primary remaining concern comes from one reviewer who viewed the contribution as overly incremental due to reliance on OSWorld task definitions and questioned whether the findings were largely intuitive. After considering the full discussion, I view this concern as partially addressed: while OSWorld-MCP does inherit tasks for comparability, the novelty lies in the evaluation setting, integrated decision-making requirement, and tool-centric analysis, which go beyond a superficial extension. The rebuttal’s added analyses and statistical evidence mitigate the “intuitive findings” critique, though the benchmark’s long-term value will indeed depend on broader MCP adoption.

Besides, in AC's opinion, the authors may discuss the recent introduction of coding actions to CUAs in some concurrent works, which somewhat offset the contributions of this paper a bit. However, this is not a weakness as they are concurrent to this submission.

**Reviewer Scores:**

- Reviewer 8XpA: No change; already positive and satisfied after rebuttal.
- Reviewer cXcb: No change; maintained a marginal-to-positive stance with concerns largely addressed.
- Reviewer DEYi: No change.
- Reviewer SQsy: Likely a modest upward revision, given that several factual and methodological concerns were explicitly addressed in the rebuttal, though a full reversal is unlikely.

---

### Decision · Program_Chairs · 2026-01-26

Accept (Poster)